# On the numerical reliability of nonsmooth autodiff: a Max-Pool case study

**Ryan Boustany**                                                                 *ryan.boustany@ut-capitole.fr*
*Toulouse School of Economics*
*Université de Toulouse*
*Thales LAS France*

**Reviewed on OpenReview:** *https://openreview.net/forum?id=142xsInVfp*

## Abstract

This paper considers the reliability of automatic differentiation for neural networks involving the nonsmooth MaxPool operation across various precision levels (16, 32, 64 bits), architectures (LeNet, VGG, ResNet), and datasets (MNIST, CIFAR10, SVHN, ImageNet). Although AD can be incorrect, recent research has shown that it coincides with the derivative almost everywhere, even in the presence of nonsmooth operations. On the other hand, in practice, AD operates with floating-point numbers, and there is, therefore, a need to explore subsets on which AD can be *numerically* incorrect. Recently, Bertoin et al. (2021) empirically studied how the choice of $\text{ReLU}'(0)$ changes the output of AD and define a numerical bifurcation zone where using $\text{ReLU}'(0) = 0$ differs from using $\text{ReLU}'(0) = 1$. To extend this for a broader class of nonsmooth operations, we propose a new numerical bifurcation zone (where AD is incorrect over real numbers) and define a compensation zone (where AD is incorrect over floating-point numbers but correct over reals). Using SGD for training, we found that nonsmooth MaxPool Jacobians with lower norms maintain stable and efficient test accuracy, while higher norms can result in instability and decreased performance. We can use batch normalization, Adam-like optimizers, or increase precision to reduce MaxPool Jacobians influence.

## 1 Introduction

Nonsmooth neural networks are trained using optimization algorithms (Bottou et al., 2018; Davis et al., 2018) based on backpropagation and automatic differentiation (AD) (Speelpenning, 1980; Rumelhart et al., 1986b; Baydin et al., 2018). AD is a crucial tool in contemporary learning architectures as it allows for fast differentiation (Griewank & Walther, 2008; Bolte et al., 2022). It is implemented in popular machine learning libraries such as TensorFlow (Abadi et al., 2016), PyTorch (Paszke et al., 2019), and Jax (Bradbury et al., 2018). Although the validity domain of AD is theoretically limited to smooth functions (Griewank & Walther, 2008), it is commonly used for nonsmooth functions (Bolte et al., 2022; 2021b; Bertoin et al., 2021). The behavior of nonsmooth AD has been investigated in previous studies (Griewank & Walther, 2008; Griewank, 2013; Griewank et al., 2016; Barton et al., 2018; Kakade & Lee, 2018; Griewank & Rojas, 2019; Griewank & Walther, 2020; Bolte & Pauwels, 2020a; Bolte et al., 2022).

**MaxPool: a nonsmooth operation**  The MaxPool operation, introduced by Yamaguchi et al. (1990), is commonly used in convolutional neural networks (CNN) for image classification (Krizhevsky & Hinton, 2010; Krizhevsky et al., 2012; Zeiler & Fergus, 2014; LeCun et al., 2015). MaxPool reduces the spatial dimensions of a feature map by selecting the maximum value within specific patches. However, when applied to uniform pixel values, MaxPool can cause nonsmoothness, especially at image edges where identical pixels can be chosen arbitrarily. In such cases, different choices of MaxPool's nonsmooth Jacobians bear a variational sense. See Appendix A.2 for an illustration. In this paper, the term *MaxPool-derived program* refers to a specific choice of a MaxPool nonsmooth Jacobian.

**Various types of nonsmooth AD errors:**  We carry out a PyTorch (Paszke et al., 2019) experiment to investigate the autodiff behavior of the nonsmooth max function, defined as $\max \colon x \mapsto \max_{1 \le i \le 4} x_i \in \mathbb{R}$. We implement two max programs ($\max_1$ and $\max_2$) with different derivative implementations. For example, the max function is not differentiable at $x = (1, 1, 1, 1)$ and autodiff returns $(1, 0, 0, 0)$ for $\max_1$ and $(0.25, 0.25, 0.25, 0.25)$ for $\max_2$ (see Appendix A.1 for more details). Let zero be a program as follows: $\text{zero} \colon t \mapsto \max_1(t \times x) - \max_2(t \times x)$. The AD output of zero is denoted by $\text{zero}'$. As mathematical functions, both $\max_1$ and $\max_2$ output the same value and zero always outputs 0. However, we observe an unexpected behavior when using AD and floating-point numbers: $\text{zero}'(t) \ne 0$ for some $t \in \mathbb{R}$.

Table 1: Overview of numerical AD errors for the zero program with 32 bits precision.

| | $\text{zero}'(t)$ | | | | | | |
|---|---|---|---|---|---|---|---|
| $t$ | $-10^{-3}$ | $-10^{-2}$ | $-10^{-1}$ | $0$ | $10^1$ | $10^2$ | $10^3$ |
| $x_1 = (1, 2, 3, 4)$ | 0.0 | 0.0 | 0.0 | $-1.5$ | 0.0 | 0.0 | 0.0 |
| $x_2 = (1.4, 1.4, 1.4, 1.4)$ | $10^{-7}$ | $10^{-7}$ | $10^{-7}$ | $10^{-7}$ | $10^{-7}$ | $10^{-7}$ | $10^{-7}$ |

For $x_1$, we observe a significant error at $t = 0$ where the computed derivative, $\text{zero}'(0)$, is $-1.5$, deviating from the correct derivative value. In contrast, for $x_2$, which often appears in tasks such as image classification (refer to Appendix A.2), theoretical calculations predict $\text{zero}'(t) = 0$ for any $t \in \mathbb{R}$. However, discrepancies emerge when using floating-point arithmetic, as illustrated by AD results. Specifically, across all $t$ values listed in Table 1, $\text{zero}'(t)$ approximates to $5.96 \times 10^{-8}$ (rounded to $10^{-7}$ in the table), which is near the computational precision limit of 32-bit systems. This phenomenon occurs due to numerical arithmetic limits. Typically, $t$ denotes a neural network parameter and $x$ represents an input image pixel area where the pixel values are identical, such as in the MNIST dataset (see Appendix A.2). The behavior observed in Table 1 are not due to the nonsmooth multivariate nature of the max function. Similar phenomena can also be seen when the univariate ReLU operation is used to compute max. In contrast, we observe no errors near machine precision using NormPool—a nonsmooth multivariate function that computes the Euclidean norm. Furthermore, reproducing Table 1 with $\text{zero} \colon t \mapsto \text{ReLU}_1(t) - \text{ReLU}_2(t)$ where $\text{ReLU}_1'(0) = 0$ and $\text{ReLU}_2'(0) = 1$, also shows no AD errors near machine precision. These observations suggest that minor AD errors may stem from the intrinsic properties of the max function. Thus, our study primarily examines max and MaxPool operations. For more details, please refer to Appendix A.3 and Appendix A.4.

**Reals vs floating-point numbers:**  Over the real numbers, AD computes derivatives for nondifferentiable functions, except on a Lebesgue measure-zero subset of inputs (Bolte & Pauwels, 2020a;b).

However, as indicated in Table 1, the use of floating-point arithmetic can expand the subsets where AD yields incorrect results. In Section 3, we propose two subsets of network parameters where AD numerically fails: a new *bifurcation zone*, characterized by significant AD amplitude variations, and a *compensation zone*, where minor amplitude variations occur near machine precision due to rounding schemes in inexact arithmetic over reals (e.g., non-associativity). Our experiments show that in a 64-bit network using MaxPool, the compensation zone covers the entire parameter space. In a 32-bit network, both compensation and bifurcation zones exist, while in a 16-bit setting, the bifurcation zone dominates the entire parameter space.

**Implications for learning dynamics:** In Section 4, we examine the influence of different nonsmooth MaxPool Jacobians on learning processes. We find that nonsmooth Jacobians with low norms produce test accuracies comparable. Conversely, Jacobians with high norms tend to reduce test accuracy, primarily due to training instability or backprop-related issues. In a 16-bit precision setting, which is a significant focus of research (Vanhoucke et al., 2011; Hwang & Sung, 2014; Courbariaux et al., 2015; Gupta et al., 2015),the effects of these Jacobians are more marked and vary depending on the network architecture, dataset, and the precision level employed. Additionally, we observe that the inclusion of batch normalization (Ioffe & Szegedy, 2015) and the use of the Adam optimizer (Kingma & Ba, 2014) help to alleviate these negative impacts. All experiments were conducted using PyTorch (Paszke et al., 2019), and our source code is available publicly [1].

**Related works and contributions:** Recent works indicate that for a broad class of programs employing nonsmooth functions, AD is incorrect primarily on a Lebesgue measure-zero subset of the program's input domain (Bolte & Pauwels, 2020a; Lee et al., 2020). However, practical inputs are typically machine-representable. In this context, Lee et al. (2023) investigated AD correctness in neural networks with machine-representable parameters, specifically excluding networks with MaxPool. Additionally, Bertoin et al. (2021) explored the impact of the $\text{ReLU}'(0)$ on AD and training, identifying a *bifurcation zone* in ReLU networks where AD is incorrect. These studies, however, do not address situations where AD is incorrect over floating-point numbers yet correct over real numbers, such as those noted in the last line of Table 1. To bridge this gap, we propose a *new bifurcation zone* and introduce the concept of a *compensation zone*. Our research evaluates the reliability of automatic differentiation in MaxPool neural networks across various precision levels, examining how the effects of the compensation zone vary with network architecture, independent of the nonsmooth functions' dimensionality (whether univariate or multivariate). We also delve into how nonsmooth MaxPool Jacobians influence the stability and performance of neural network training.

**Organization of the paper:** In Section 2, we discuss the elements of nonsmooth backpropagation and define three subsets of network parameters - the bifurcation zone, compensation zone, and regular zone. We also examine the implications of nonsmooth MaxPool Jacobians for backpropagation, based on Bolte & Pauwels (2020a;b). In Section 3, we define and discuss the numerical bifurcation and compensation zones, including factors that influence their significance. In Section 4, we present detailed experiments on neural network training. For additional findings and experiments, please refer to Appendix C.

---

[1] https://github.com/ryanboustany/MaxPool-numerical

## 2 MaxPool neural networks and nonsmooth AD

### 2.1 Preliminaries and notations

In supervised training for neural networks, we work with a set of training data $(x_i, y_i)_{i=1}^N$, where each $x_i$ is an input and $y_i$ its matching label. A neural network, through its function $f$, uses parameters $\theta$ to generate predictions $\hat{y}_i = f(x_i, \theta)$. The difference between these predictions and the actual labels is measured by a loss function $\ell$. The aim is to reduce this discrepancy across the training set by minimizing an empirical loss function $L$ such as:

$$\min_{\theta \in \mathbb{R}^p} \quad L(\theta) := \frac{1}{N} \sum_{i=1}^N \ell(\hat{y}_i, y_i). \tag{1}$$

For all $i \in \{1, \ldots, N\}$ and $\theta \in \mathbb{R}^p$, Equation (1) can be expressed with $\ell(\hat{y}_i, y_i) = l_i(\theta)$, where $l_i : \mathbb{R}^p \to \mathbb{R}$ represents a composition of $H$ elementary functions as follows:

$$l_i(\theta) = g_{i,H} \circ g_{i,H-1} \circ \ldots \circ g_{i,1}(\theta). \tag{2}$$

Equation (2) models common neural network types, including feed-forward (Rumelhart et al., 1986a), convolutional (LeCun et al., 1998), and recurrent networks (Hochreiter & Schmidhuber, 1997). For a more concrete example, please refer to Appendix A.2 in Bertoin et al. (2021). We focus on elementary functions that are locally Lipschitz and semialgebraic, commonly found in nonsmooth neural networks (Bolte & Pauwels, 2020a;b). Functions $g_{i,j}$ include operations such as linear transformations, ReLU, MaxPool, convolution with filters, and softmax for classification.

### 2.2 Nonsmooth AD framework

Training nonsmooth neural networks (Bolte & Pauwels, 2020a; Bolte et al., 2021b; 2022; 2021a; Davis et al., 2020) is challenging due to the need to compute subgradients from Equation (1). Major machine learning tools such as TensorFlow (Abadi et al., 2016), PyTorch (Paszke et al., 2019), and Jax (Bradbury et al., 2018) address this issue using automatic differentiation, referred to here as backprop (Rumelhart et al., 1986b; Baydin et al., 2018). They apply differential calculus to nonsmooth items, often replacing derivatives with *Clarke Jacobians* (Clarke, 1983). Given a locally Lipschitz continuous function $F : \mathbb{R}^p \to \mathbb{R}^q$, the *Clarke Jacobian* of $F$ is defined as:

$$\text{Jac}^c F(x) = \text{conv} \left\{ \lim_{k \to +\infty} \text{Jac} F(x_k) : x_k \in \text{diff}_F, x_k \underset{k \to +\infty}{\to} x \right\} \tag{3}$$

where $\text{diff}_F$ represents the full measure set where $F$ is differentiable and $\text{Jac} F$ is the standard Jacobian of $F$. A selection $v$ in $\text{Jac}^c F$ is a function $v : \mathbb{R}^p \to \mathbb{R}^{p \times q}$ such that, for all $x \in \mathbb{R}^p$, $v(x) \in \text{Jac}^c F(x)$. If $F$ is $C^1$, the only possible selection is $v = \text{Jac} F$.

**Definition 1 (Calculus model, programs and nonsmooth AD)** Let $l$ be a composition function evaluated at $\theta \in \mathbb{R}^p$, as specified in Equation (2). A program $P$ that executes $l$ can be described through a sequence of subprograms such as:

- Elementary programs: $\{g_j\}_{j=1}^H$ such that $l(\theta) = g_H \circ g_{H-1} \circ \ldots \circ g_1(\theta)$.

- Derived programs: $\{v_j\}_{j=1}^H$ where each $v_j(w) \in \text{Jac}^c g_j(w)$ at point $w = g_{j-1} \circ \cdots \circ g_1(\theta)$.

Then, the backprop algorithm automates applying differential calculus rules as follows:

$$\text{backprop}[P](\theta) = v_H\left(g_{H-1} \circ \ldots \circ g_1(\theta)\right) \cdot v_{H-1}\left(g_{H-2} \circ \ldots \circ g_1(\theta)\right) \cdot \ldots \cdot v_1(\theta). \qquad (4)$$

In practice, AD libraries (Abadi et al., 2016; Paszke et al., 2019; Bradbury et al., 2018) implement dictionaries (see for e.g. Griewank & Walther (2008); Bolte et al. (2022)) containing conjointly elementary programs and derived programs which efficiently computes the quantities defined in Equation (4).

**Remark 1** As seen in Section 1 with the zero program, various programs can implement a unique composition function $l$. Each nonsmooth elementary program $g_j$ in the composition (see Definition 1) can be associated with different derived programs $v_j$. Specifically, for any $j = 1, \ldots, H$ and $w = g_{j-1} \circ \cdots \circ g_1(\theta)$, all selections $v_j(w)$ from the Clarke Jacobian of $g_j(w)$ bear a variational sense.

**Example 1** The Clarke subdifferential of $\text{ReLU}(t) = \max(0, t)$ at $t$ is 0 for $t < 0$, 1 for $t > 0$, and the interval $[0, 1]$ for $t = 0$. All ReLU-derived program that implements $\text{ReLU}'(0) = s$ with $s \in [0, 1]$ can be used for backprop.

**Definition 2 (Backprop set)** Let $l$ denote a composition function evaluated at $\theta \in \mathbb{R}^p$, as specified in Equation (2). We define $J(\theta)$ as the function that encompasses the set of all possible backprop outputs through all programs implementing $l(\theta)$ as in Definition 1:

$$J(\theta) = \{\text{backprop}[P](\theta) : P \text{ is a program implementing } l(\theta)\}. \qquad (5)$$

**Remark 2** For a composition function $l$ composed by $C^1$ elementary programs $\{g_j\}_{j=1}^H$, $J(\theta)$ is a singelton for all $\theta \in \mathbb{R}^p$. For locally Lipchitz semialgebraic (or definable) elementary programs $\{g_j\}_{j=1}^H$: Equation (4) returns an element within the backprop set.

**Remark 3** The chain rule, essential for AD, often fails with Clarke subgradients. Hence, the backprop set might differ from the Clarke subdifferential (Clarke, 1983). For example, the Clarke subdifferential of $2\text{ReLU}(x) - \frac{1}{3}\text{ReLU}(-x)$ at $x = 0$ is $[\frac{1}{3}, 2]$, whereas backprop outputs 0 (with $\text{ReLU}'(0) = 0$).

## 2.3 Network parameters subsets

Recently, Bertoin et al. (2021) studied the bifurcation zone in ReLU networks, characterized by network parameters at which the output of AD diverges between $\text{ReLU}'(0) = 0$ and $\text{ReLU}'(0) = 1$. However, their study did not cover the scenario where backprop theoretically computes a singleton, but AD inaccurately computes it due to floating-point arithmetic. From our knowledge, this phenomenon appears when comparing the output of AD across different derivatives implementations of MaxPool. To address this gap, we introduce the concept of the compensation zone.

**Definition 3 (Compensation, bifurcation and regular zones)** For each $i = 1, \ldots, N$, let $l_i$ denote a composition function evaluated at $\theta \in \mathbb{R}^p$ and $J_i(\theta)$ denote the backprop set associated as detailed in Definition 2. We define the following network parameters subsets of $\mathbb{R}^p$:

$$\Theta_R = \left\{\theta \in \mathbb{R}^P : \forall i, j \in \{1, \ldots, N\} \times \{1, \ldots, H\}, \text{Jac}^c\, g_{i,j}(w) \text{ is a singleton}\right\}, \qquad (6)$$

$$\Theta_C = \left\{\theta \in \mathbb{R}^P \backslash \Theta_R : \forall i \in \{1, \ldots, N\}, J_i(\theta) \text{ is a singleton}\right\}, \qquad (7)$$

$$\Theta_B = \left\{\theta \in \mathbb{R}^p \backslash \Theta_R : \exists i \in \{1, \ldots, N\} \text{ such that } J_i(\theta) \text{ is not a singleton}\right\}, \qquad (8)$$

where $w = g_{i,j-1} \circ \ldots \circ g_{i,1}(\theta)$, $\Theta_R$ is the regular zone, $\Theta_C$ the compensation zone and $\Theta_B$ the bifurcation zone.

The mathematical tools of Proposition 1 are conservative fields developed in Bolte & Pauwels (2020a). This proposition implies that theoretically (assuming exact arithmetic over the reals), the backprop set is almost everywhere a singleton. The proof is given in Appendix B.

**Proposition 1** Given subsets $\Theta_R$, $\Theta_B$, and $\Theta_C$ in $\mathbb{R}^p$ as defined in Definition 3, the following properties hold:

- $\Theta_R$, $\Theta_B$, and $\Theta_C$ form a partition of $\mathbb{R}^p$.

- $\Theta_B$ is a Lebesgue null measure subset.

**Remark 4 (Backprop returns a gradient a.e.)** Let $\theta \in \mathbb{R}^p$ and $P$ be a program implementing a composition function $l(\theta)$ as in Definition 1. Then $\text{backprop}[P](\theta) = \nabla l(\theta)$ almost everywhere.

### 2.4 MaxPool-derived programs

**Definition 4 (Clarke Jacobian of matrix's maximum function)** Let $X$ be a $m \times n$ real matrix and $F_s$ be a function such that $F_s(X) = \max_{1 \le i \le m, 1 \le j \le n} X_{ij} \in \mathbb{R}$, where $s := m \times n$ denotes the size of $X$. The Clarke Jacobian of $F_s$ at the point $X$ is:

$$\text{Jac}^c F_s(X) = \text{conv}\left( \bigcup_{(i,j) \in A(X)} E_{ij} \right), \tag{9}$$

where $A(X) := \{(i,j) \in \{1,\ldots,m\} \times \{1,\ldots,n\} : F_s(X) = X_{ij}\}$ is the active set and $E_{ij}$ is an $m \times n$ matrix with all entries equal to 0 except for the $(i,j)$-th entry which is 1.

**Definition 5 (MaxPool operation)** Let $X \in \mathbb{R}^{p \times q}$ be a real matrix, and $s := m \times n$ be the size of a pooling window such that $p \ge m$ and $q \ge n$. For each $i \in \{0, \ldots, \lfloor \frac{p}{m} \rfloor - 1\}$ and $j \in \{0, \ldots, \lfloor \frac{q}{n} \rfloor - 1\}$, we define a submatrix $X_{i,j}$ of $X$, of size $m \times n$ as follows:

$$X_{i,j} := \{X_{kl} : m \times i \le k < m \times (i+1), n \times j \le l < n \times (j+1)\}, \tag{10}$$

where $k$ and $l$ are the indices of the entries in $X$, in the lexicographic order. The MaxPool operation output a matrix $Y \in \mathbb{R}^{\lfloor \frac{p}{m} \rfloor \times \lfloor \frac{q}{n} \rfloor}$ where $Y_{ij} = F_s(X_{i,j})$ for all $i \in \{0, \ldots, \lfloor \frac{p}{m} \rfloor - 1\}$ and $j \in \{0, \ldots, \lfloor \frac{q}{n} \rfloor - 1\}$. Finally, the MaxPool Clarke Jacobian at point $X$, denoted as $\text{Jac}^c \text{MaxPool}(X)$, can be obtained by replacing each submatrix $X_{i,j}$ in $X$ with $\text{Jac}^c F_s(X_{i,j})$.

**Definition 6 (MaxPool-derived programs)** Define $X_{i,j} \in \mathbb{R}^{m \times n}$ as a submatrix of $X$ (Definition 5), from which we derive MaxPool programs based on the Clarke Jacobian:

- **Native:** Chooses the first index $(i_1, j_1)$ from the active set $A(X_{i,j})$ and outputs $E_{i_1 j_1}$. Autograd libraries use this implementation.

- **Minimal:** Takes all indices from $A(X_{i,j})$, averaging them as $\frac{1}{|A(X_{i,j})|} \sum_{(k,l) \in A(X_{i,j})} E_{kl}$. We called it "minimal" as it yields the smallest norm element within Equation (9).

- **Hybrid:** A blend of native and minimal, parameterized by $\beta > 0$:

$$(1 - \beta) \cdot E_{i_1 j_1} + \beta \cdot \left( \frac{1}{|A(X_{i,j})|} \sum_{(k,l) \in A(X_{i,j})} E_{kl} \right),$$

**Remark 5** The hybrid MaxPool-derived program is a selection of the MaxPool Clarke Jacobian for $\beta \in [0,1]$ and a selection of a conservative Jacobian approach for other $\beta$ values, as outlined in Bolte & Pauwels (2020a). .

The chain rule is essential for AD, but it often faces challenges with Clarke subgradients. The backprop set defined in Definition 2 does not consistently qualify as a Clarke subdifferential. Therefore, in the following sections of this paper, we will also investigate the impact of using a conservative Jacobian ($\beta > 1$) on the learning and training processes. This investigation is important as conservative Jacobians have a variational interpretation within the framework of nonsmooth AD (see Bolte & Pauwels (2020a)).

## 3 A more general numerical bifurcation zone

In this section, we explore numerically subsets of network parameters for neural networks involving MaxPool operations across various floating-point precisions. We find that the numerical bifurcation zone identified by Bertoin et al. (2021) does not apply to scenarios involving MaxPool-derived programs. Specifically, both the max function and MaxPool contribute to minor AD errors that appear with floating-point numbers but not with real numbers. As a result, we suggest a new numerical bifurcation and define a compensation zone using two distinct methods: one with nondeterministic GPU computations and the other with ReLU-derived programs. These methods are based on the notations and frameworks discussed in Sections 2.1 and 2.2.

### 3.1 A numerical criteria for the bifurcation and compensation zone

**Numerical bifurcation zone for** ReLU **networks:** Recently, Bertoin et al. (2021) investigated a numerical bifurcation zone $S_{01}$ specific to ReLU-derived programs. For each $i = 1, \ldots, N$, two programs implement a same function $l_i$: $R_i^0$ (using ReLU$'(0) = 0$) and $R_i^1$ (using ReLU$'(0) = 1$). The bifurcation zone $S_{01}$ is defined as:

$$S_{01} = \left\{ \theta \in \mathbb{R}^P : \ \exists i \in \{1, \ldots, N\}, \text{backprop}[R_i^0](\theta) \neq \text{backprop}[R_i^1](\theta) \right\}. \tag{11}$$

**Definition 7 (Backprop variation)** Let $(B_q)_{q \in \mathbb{N}}$ be a sequence of mini-batches, where each batch size $|B_q|$ falls within $\{1, \ldots, N\}$. Consider $P = \{P_i\}_{i=1}^N$ and $Q = \{Q_i\}_{i=1}^N$ as two neural network implementations using different nonsmooth-derived programs (e.g., ReLU or MaxPool). *Each $P_i$ and $Q_i$ computes a composition function $l_i$. The backprop variation between $P$ and $Q$ over $M$ experiments with random parameters $\{\theta_m\}_{m=1}^M$ is defined as:*

$$D_{m,q}(P,Q) = \left\| \text{backprop} \left[ \sum_{i \in B_q} P_i(\theta_m) \right] - \text{backprop} \left[ \sum_{i \in B_q} Q_i(\theta_m) \right] \right\|_1. \tag{12}$$

**A 32 bits MNIST experiment:** Let $P$ and $Q$ be programs for a LeNet-5 network on MNIST, using native and minimal MaxPool programs, respectively. For a sanity check, let $\tilde{P}$ be a copy of $P$. We compute the backprop variation (as in Definition 7) between $P$ and $\tilde{P}$ and between $P$ and $Q$. We control all sources of divergence in our implementation using deterministic computation. Results are reported in Figure 1 and the experiment was run on a CPU under 32 bits precision.

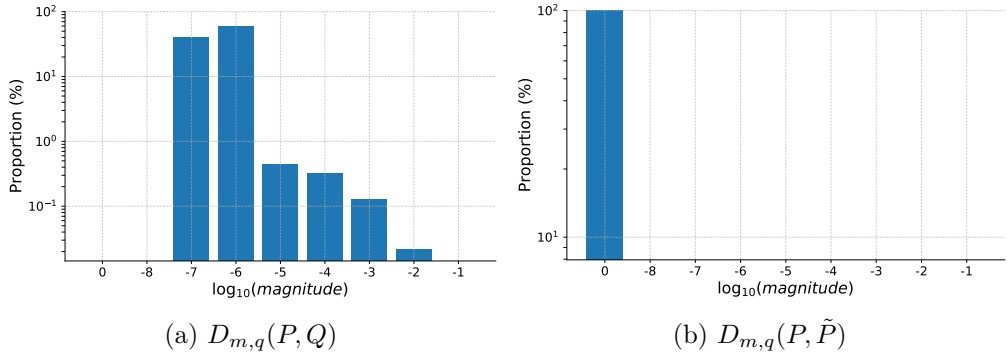

(a) $D_{m,q}(P, Q)$         (b) $D_{m,q}(P, \tilde{P})$

Figure 1: Histogram of backprop variation $D_{m,q}$ for LeNet-5 on MNIST (128 mini-batch size) at 32-bit precision, comparing $P$ with $\tilde{P}$ and $P$ with $Q$ over $M = 1000$ experiments.

We observe no variation in backpropagation between $P$ and $\tilde{P}$, indicating controlled sources of divergence. This observation contrasts with the expectations from Proposition 1, which predicts no variation between $P$ and $Q$; we find $D_{m,q}(P, Q) > 0$ across all $m, q$. We identify two types of variations: minor ones, comprising 98.78% of parameters, which align with machine precision in 32 bits (between $10^{-8}$ and $10^{-7}$), and major ones, peaking at $10^{-3}$ and accounting for 1.22% of parameters. Notably, these findings differ significantly from the backprop variations observed with ReLU-derived programs as shown in Figure 3, where we either see significant divergences or none. Consequently, this section focuses on analyzing backprop variations in MaxPool-derived programs to propose a new bifurcation zone.

**An heuristic for the numerical bifurcation zone:** In Figure 1, we identify two types of backpropagation variations: one potentially arising from numerical bifurcations and another due to floating-point arithmetic errors, which we refer to as compensation errors. To establish criteria for proposing a new numerical bifurcation zone for nonsmooth-derived programs, we compare these observed backpropagation variations with known sources, such as GPU nondeterminism and variations from ReLU-derived programs at 16 and 32-bit precision. This method allows us to differentiate between numerical bifurcations and compensation errors without presuming distinct zones. We denote floating-point precision by $\omega$ and consider various neural networks like LeNet-5, VGG, or ResNet for our analysis.

**A threshold with nondeterministic GPU computations:** We set a threshold $\tau_{f,\omega}^1$ for the maximum backprop variation due to nondeterministic GPU computations (refer to Appendix A.6.1 for more details):

$$\tau_{f,\omega}^1 = \max_{1 \leq m \leq M, 1 \leq q} D_{m,q}(P, \tilde{P}) \tag{13}$$

where $P$ and $\tilde{P}$ compute a neural network $f$ using the same nonsmooth-derived program, for example, $\text{ReLU}'(0) = 0$ or minimal MaxPool. See Figure 2 for an illustration. We observe no variation at $\omega = 16$, with PyTorch's nondeterministic GPU operations disabled. Additionally, $\tau^1$

can be interpreted as an upper bound on the expected backprop variation when repeatedly running the same program under nondeterministic conditions.

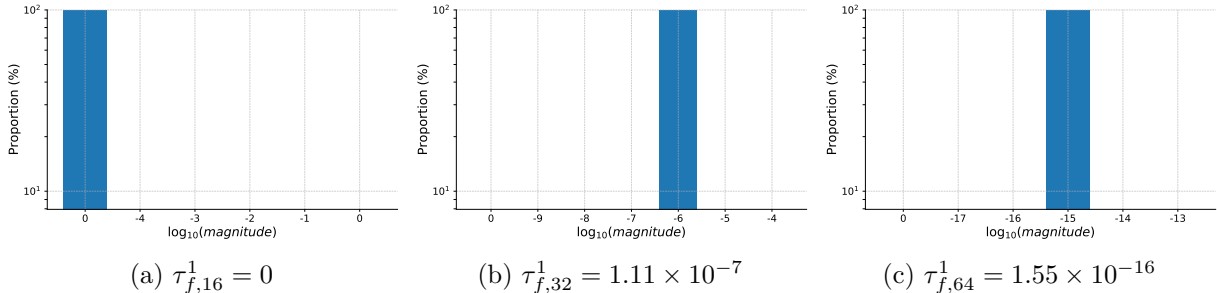

(a) $\tau^1_{f,16} = 0$      (b) $\tau^1_{f,32} = 1.11 \times 10^{-7}$      (c) $\tau^1_{f,64} = 1.55 \times 10^{-16}$

Figure 2: Histogram of backprop variation under nondeterministic GPU operations, where $f$ is a LeNet-5 network on MNIST with batch size 128 for $M = 1000$ experiments.

**A threshold with ReLU-derived programs:** For ReLU-derived programs, we define $R^0$ (with $\text{ReLU}'(0) = 0$) and $R^1$ (with $\text{ReLU}'(0) = 1$) as two programs implementing a same network $f$ under deterministic GPU operations. We introduce threshold $\tau^2_{f,\omega}$ for backprop variation:

$$\tau^2_{f,\omega} = \min_{1 \leq m \leq M, 1 \leq q} \left\{ D_{m,q}(R^0, R^1) : D_{m,q}(R^0, R^1) > 0 \right\}, \tag{14}$$

$\tau^2$ can be interpreted as a lower bound on the error we expect to make when running two different ReLU-derived programs of the same function.

**Remark 6** We do not consider $\tau^2$ in the context of MaxPool-derived programs, as it is anticipated that $\tau_2$ will approximate machine precision values as in Figure 1.

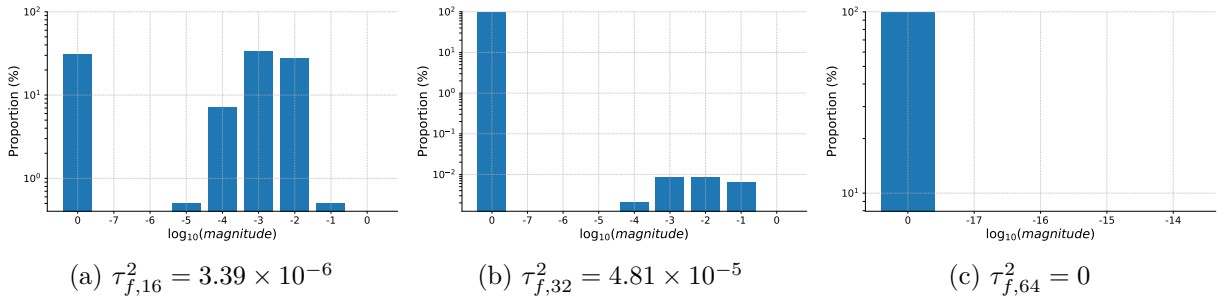

(a) $\tau^2_{f,16} = 3.39 \times 10^{-6}$      (b) $\tau^2_{f,32} = 4.81 \times 10^{-5}$      (c) $\tau^2_{f,64} = 0$

Figure 3: Histogram of backprop variation with ReLU-derived programs, where $f$ is a LeNet-5 network on MNIST with batch size 128 for $M = 1000$ experiments.

Figure 3 illustrates two types of backpropagation variations with ReLU-derived programs: significant divergences or none at all, contrasting with phenomena observed with MaxPool-derived programs. These divergences may suggest the presence of a numerical bifurcation zone. Additionally, variations from nondeterministic GPU computations, as shown in Figure 2, correspond to minor variations near machine precision, similar to those seen in Figure 1. We propose establishing a numerical bifurcation zone, applying different thresholds for various precisions to accommodate hardware constraints.

**Criteria 1 (Numerical bifurcation zone)** For a neural network $f$ and a floating-point precision $\omega$, let $\tau_{f,\omega}$ be a fixed threshold (for e.g $\tau_{f,\omega}^1$, $\tau_{f,\omega}^2$). The numerical bifurcation zone is defined as:

$$S(\tau_{f,\omega}) = \left\{ \theta \in \mathbb{R}^P : \exists i \in \{1, \ldots, N\}, \|\text{backprop}[P_i](\theta) - \text{backprop}[Q_i](\theta)\|_1 > \tau_{f,\omega} \right\} \subset \Theta_B. \quad (15)$$

Here, $P_i$ and $Q_i$ are programs implementing $f$ using different nonsmooth-derived programs.

Table 4 in Appendix A.6 lists threshold values for various networks and datasets across 16-bit, 32-bit, and 64-bit precisions. These thresholds are numerical guides and fluctuate based on the initial network parameters, datasets, and architecture. The characteristics of the compensation zone depend on the neural network's structure rather than the nature (univariate or multivariate) of the nonsmooth elementary programs defined in Definition 1. For example, computing MaxPool using ReLU programs can lead to compensation errors, as detailed in Table 1 (see Appendix A.3). In convolutional networks like VGG or ResNet, computing MaxPool with ReLU functions using the formula $2 \max(x, y) = (x + y) + (\text{ReLU}(x) - \text{ReLU}(-y)) + (\text{ReLU}(y) - \text{ReLU}(-x))$ does not align with the bifurcation zone proposed by Bertoin et al. (2021). Conversely, using NormPool—a nonsmooth multivariate function calculating the Euclidean norm—avoids such compensation errors. Further details can be found in Appendix A.4.

## 3.2 Volume of the numerical bifurcation zone

We employed Monte Carlo sampling to estimate the volume of the numerical bifurcation zone for various networks, adhering to Criteria 1. Thresholds $\tau_{f,16}^2$, $\tau_{f,32}^1$, and $\tau_{f,64}^1$ were consistently applied across all networks, as detailed in Appendix A.6.3 with reference to Equations (13) and (14).

**Experimental Setup:** We generated a set of network parameters $\{\theta_m\}_{m=1}^M$ randomly using Kaiming-Uniform initialization (He et al., 2015), with $M = 1000$ experiments conducted. Subsequently, we iterated over the entire CIFAR10 dataset to estimate the proportion of $\theta_m$ within the numerical bifurcation zone $S$ as defined in Criteria 1 (referenced in Equation (18)) and the proportion of affected mini-batches (detailed in Equation (19)).

**Impact of floating-point precision:** Using the VGG11 model on the CIFAR10 dataset, we evaluated the volume of $S$ across different precision levels. The results revealed that at 16-bit and 32-bit precision, all parameters resided within $S$, whereas at 64-bit precision, none did. This variance demonstrates the significant role of precision in the effects of backprop with MaxPool-derived programs. Notably, the impact on mini-batches was substantial, with 46% at 32 bits and 100% at 16 bits, underscoring the influence of precision on the computational outcomes.

Table 2: Impact of $S$ according to floating-point precision using a VGG11, on CIFAR10 dataset and $M = 1000$ experiments. The first line represents network parameters $\theta_m$ in $S$, while the second measured the proportion of affected mini-batches falling in $S$.

| Floating-point precision | 16 bits | 32 bits | 64 bits |
|---|---|---|---|
| Proportion of $\{\theta_m\}_{m=1}^M$ in $S$ | 100% | 100% | 0% |
| Proportion of impacted mini-batches | 100% | 46.67% | 0% |

We also investigated the influence of mini-batch size on the proportion of affected mini-batches within the numerical bifurcation zone $S$ using the VGG11 model on the CIFAR10 dataset. Our

findings indicate that larger mini-batch sizes correlate with an increased proportion of impacted mini-batches at 32-bit precision. However, at 64-bit precision, no parameters were observed to fall into $S$ (as shown in Figure 4). Additionally, variations in network depth—examined across VGG variants 11, 13, 16, and 19—did not significantly alter the impact on mini-batches at 16-bit and 32-bit precisions. Notably, the introduction of batch normalization markedly increased the number of affected mini-batches at 32-bit precision.

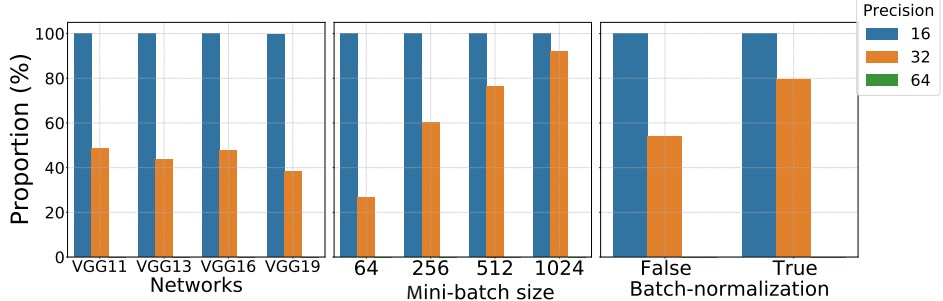

Figure 4: Impact of different size parameters on the proportion of affected mini-batches (see Equation (19) using CIFAR10 dataset. First: Different VGG network sizes. Second: VGG11 with varying mini-batch sizes. Third: VGG11 with and without batch normalization.

## 4 Impact on learning

### 4.1 Benchmarks and implementation

**Datasets and architectures:** We train neural networks to investigate the impact of numerical effects outlined in Section 3. Our experiments used CIFAR10 (Krizhevsky & Hinton, 2010), MNIST (LeCun et al., 1998) and ImageNet (Deng et al., 2009) datasets. We test various network architectures including VGG11 (Simonyan & Zisserman, 2014), ResNet (He et al., 2016), and LeNet (LeCun et al., 1998). Details are available in Appendix C.1.

**Training settings:** The default optimizer is SGD. Conducted on PyTorch and Nvidia V100 GPUs, we define mini-batch sequences $(B_q)_{q \in \mathbb{N}}$ with sizes $|B_q| \subset \{1, \ldots, N\}$, where $\alpha_q > 0$ is the learning rate for each mini-batch $q$. Each program $P_i$ in $P = \{P_i\}_{i=1}^N$ implements a function $l_i$ (as in Definition 1). The SGD algorithm updates network parameters $\theta_{q,P}$ by:

$$\theta_{q+1,P} = \theta_{q,P} - \gamma \frac{\alpha_q}{|B_q|} \sum_{i \in B_q} \text{backprop}[P_i](\theta_{q,P}) \tag{16}$$

with $\gamma > 0$ indicating the step-size parameter.

### 4.2 Effect on training and test errors

We further investigate the effect of the phenomenon described in Section 3 in terms of learning using the CIFAR10 dataset (Krizhevsky & Hinton, 2010) and the VGG11 architecture (Simonyan & Zisserman, 2014). This was performed at 16-bit and 32-bit precisions with various $\beta$ values, repeating each configuration ten times with random initialization. The results are depicted in Figure 5. To confirm our findings in alternative settings, we also use the MNIST (LeCun et al., 1998) and ImageNet (Krizhevsky et al., 2012) datasets and the ResNet18 and ResNet50 architectures (He

et al., 2016). Additional details on the different architectures and datasets experiments are found in Appendix C.

**Training effect with 16-bit:** For $\beta$ values greater than $10^3$, we observe training instability characterized by exploding gradients, unaffected by the presence of batch normalization. Conversely, stable and efficient test accuracies are maintained for $\beta$ values within the set $\{0, 1, 10, 100\}$.

**Training effect with 32-bit:** When $\beta$ values are large, such as $10^4$, the training process may become unstable, exhibiting oscillations and sudden fluctuations in the learning curve. This instability can occur if batch normalization is not used. However, incorporating batch normalization with high $\beta$ values helps stabilize the training dynamics, enhances test data accuracy, and prevents gradient explosion.

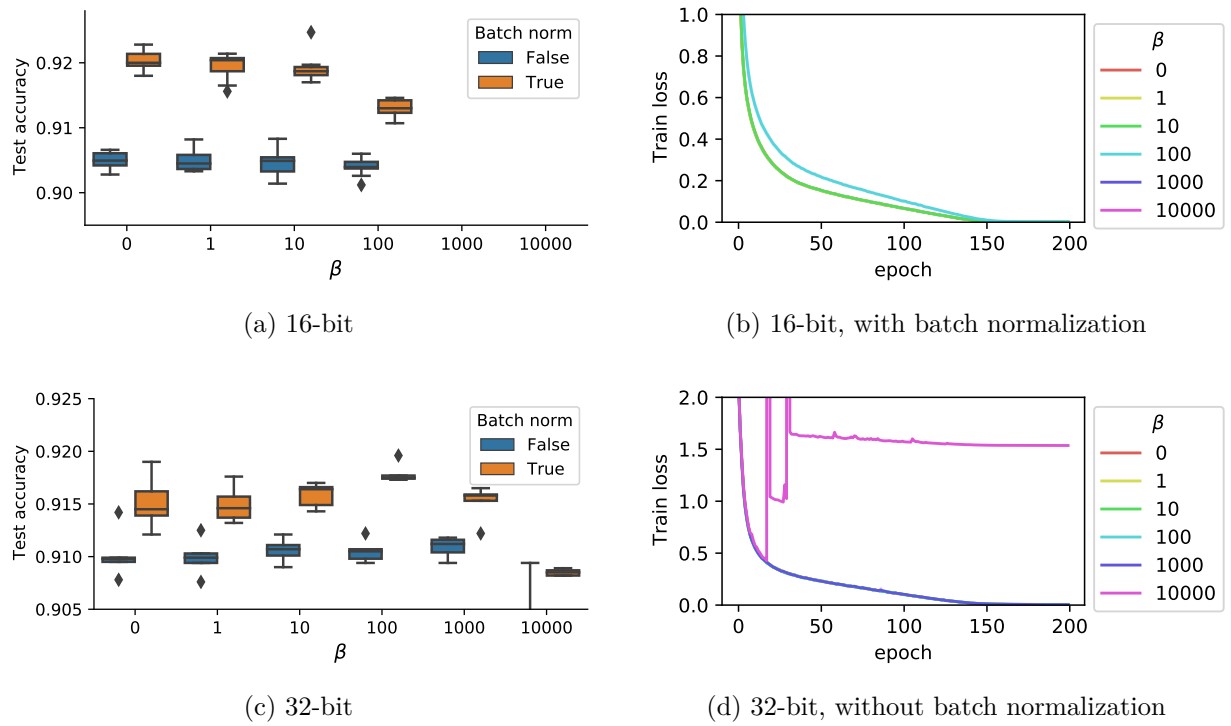

(a) 16-bit

(b) 16-bit, with batch normalization

(c) 32-bit

(d) 32-bit, without batch normalization

Figure 5: Training a VGG network on CIFAR10 with SGD. We performed ten random initializations for each experiment, depicted by the boxplots and the filled contours (standard deviation).

**Training and weight differences:** We trained seven VGG11 networks $\{P_i\}_{i=0}^{6}$ at 32-bit precision on CIFAR10 for 200 epochs, using 128-size mini-batches, fixed learning rate for each mini-batch $q$ $\alpha_q = 1.0$, and step-size parameter $\gamma \in [0.01, 0.012]$. All networks, starting with the same parameters, varied with hybrid MaxPool $\{\beta_i\}_{i=0}^{6}$. Using nondeterministic GPU computation, we measured epoch-wise backpropagation differences between $P_0$ and the others, observing parameter variations and test accuracies. Variations and accuracies for $\beta \leq 10^3$ were consistent, showing $\beta$'s minimal impact for practical scenarios. At $\beta = 10^4$, significant divergences and a test accuracy drop were noted, indicating that high $\beta$ values could destabilize training due to exploding gradients.

**Recommendation for practitioners:** We found that the choice of $\beta$ significantly influences neural network training. Specifically, large $\beta$ values can destabilize training and reduce test accuracy.

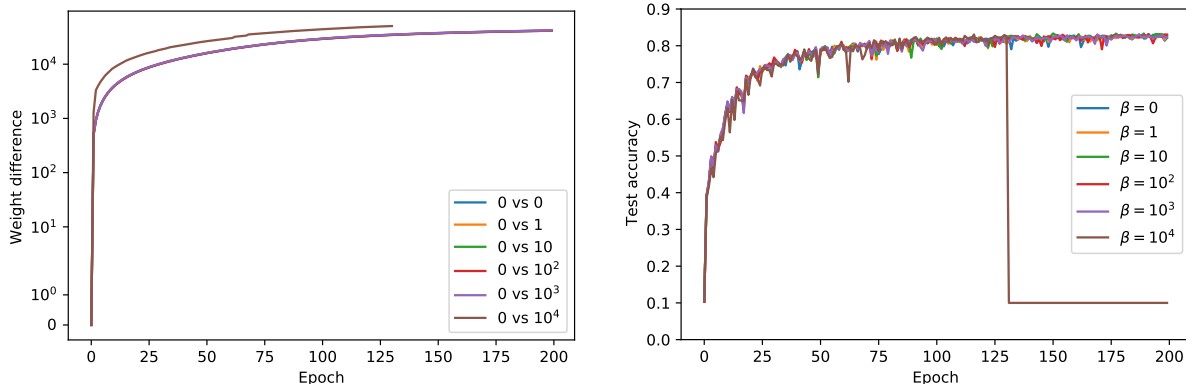

Figure 6: Left: Difference between network parameters ($L^1$ norm) at each epoch. "0 vs 0" indicates $\|\theta_{k,P_0} - \theta_{k,P_7}\|_1$ where $P_7$ is a second run of $P_0$ for sanity check, "0 vs 1" indicates $\|\theta_{k,P_0} - \theta_{k,P_1}\|_1$. Right: test accuracy of each $\{P_i\}_{i=0}^5$ during 200 epochs.

It is noteworthy that such large values are impractical in real-world applications. For more realistic $\beta$ values, we observed no impact on training loss or test accuracy. We recommend using low-norm Jacobians to ensure robust training; therefore, we propose a minimal MaxPool Jacobian ($\beta = 1$ results in the minimal norm). Furthermore, employing the Adam optimizer at 32-bit precision, as noted by Bertoin et al. (2021), effectively counteracts the adverse effects of large $\beta$ values, thereby stabilizing training (see Appendix C.2).

**Connexion with the choice of** $\text{ReLU}'(0)$**:** Initially, Bertoin et al. (2021) reported that the choice of $\text{ReLU}'(0)$ significantly impacts learning, with vanilla SGD training showing $\text{ReLU}'(0) = 0$ as the most efficient option. A recent erratum published by the same authors (Bertoin et al. (2023)) revises this finding, indicating that the impact of $\text{ReLU}'(0)$ on learning outcomes is considerably less pronounced than initially stated. This is in line with our investigation of the MaxPool operation.

## 5 Conclusion

In our study, we evaluate the reliability of automatic differentiation in neural networks involving MaxPool. Testing across a variety of models and datasets revealed that AD may inaccurately process MaxPool operations when using floating-point numbers. This observation suggests that the AD correctness findings by Lee et al. (2023) may not fully apply to convolutional neural networks incorporating MaxPool. Our analysis identifies two critical zones: bifurcation zones, where AD inaccuracies manifest in both real and floating-point computations, and compensation zones, which are accurate in real numbers but may exhibit errors in floating-point representations. Although bifurcation zones are uncommon, they lead to significant AD discrepancies. In contrast, compensation zones more frequently display minor shifts in amplitude, related to machine precision.

Furthermore, employing lower-norm MaxPool Jacobians tends to enhance training stability and test accuracy, while higher-norm Jacobians increase the risk of training instability, especially in lower-precision environments. Factors such as dataset characteristics, network architecture, and learning parameters—including batch normalization and the Adam optimizer—significantly influence AD's numerical behavior.

## Acknowledgments and Disclosure of Funding

The author acknowledges the support of the AI Interdisciplinary Institute ANITI funding under the grant agreement ANR-19-PI3A-0004. The author acknowledges the help of the Association Nationale de la Recherche et de la Technologie (ANRT) and Thales LAS France, which contributed to Ryan B's grant. This work was performed using HPC resources from CALMIP (Grant 2023-[P23040]). The author would like to thank Jérôme Bolte and Edouard Pauwels for their precious feedback. The author would like to thank the collaborators at Thales LAS France, in particular Beatrice Pesquet-Popescu and Andrei Purica, for their help. comments.

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

This is the appendix for "On the numerical reliability of nonsmooth autodiff: a MaxPool case study".

## Contents

## A    Further comments, discussion, and technical elements

### A.1    Implementation of the zero program

The implementation of the zero function used in Table 1 is given in Figure 7. Programs $\max_1$ and $\max_2$ correspond to an equivalent implementation of the same function max, but the computed derivatives are different.

```python
def max1(x):                              def max2(x):
    res = x[0]                                return torch.max(x)
    for i in range(1, 4):
        if x[i] > res:                    def zero(t):
            res = x[i]                        z = t * x
    return res                                return max1(z) - max2(z)
```

Figure 7: Implementation of programs $\max_1$, $\max_2$ and zero using Pytorch. Programs $\max_1$ and $\max_2$ are an equivalent implementation of max, but with different derivatives due to the implementation.

### A.2    Challenges posed by MaxPool in image processing

In Convolutional Neural Networks (CNNs), the MaxPool operation is frequently used for reducing dimensions and downsampling. This function is especially crucial in image contexts, where uniform intensity regions are common, especially around the edges of objects and flat surfaces. One common situation is encountering identical pixel values within a pooling window, as shown in Figure 8. MaxPool must choose among these equivalent values, creating a point of non-differentiability. During training, this affects gradient calculation in backpropagation, affecting the updates to convolutional filters (Goodfellow et al., 2016).

### A.3    AD errors with ReLU-derived programs

We conduct a small PyTorch experiment using the nonsmooth function $\mathrm{ReLU}\colon x \mapsto \max(x, 0)$. Consider two programs $\max_1$ and $\max_2$ implementing the $\max\colon x \mapsto \max_{1 \leq i \leq 4} x_i \in \mathbb{R}$ function using different ReLU-derived programs. Note that $2\max(x, y) = (x+y) + (\mathrm{ReLU}(x) - \mathrm{ReLU}(-y)) + (\mathrm{ReLU}(y) - \mathrm{ReLU}(-x))$. Let $\mathrm{zero}_2\colon t \mapsto \max_1(t \times x) - \max_2(t \times x)$ be a program implementing

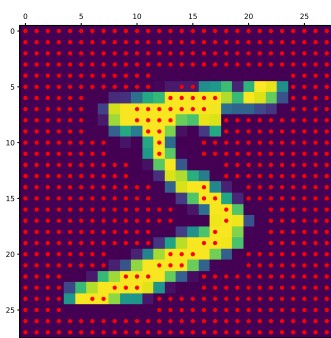

Figure 8: Image segment post-convolution, spotlighting equal pixel values (marked in red) within a 2x2 MaxPool window.

the null function as described in Figure 9. Let $\text{zero}_2'$ denote the backward AD algorithm for the zero program. As mathematical functions, $\max_1$ and $\max_2$ are equal and the program zero outputs constantly 0. However, for some $t \in \mathbb{R}$, AD can return $\text{zero}_2'(t) \neq 0$. Results are reported in Table 3 and similar to Table 1.

```python
def relu(x):
    return torch.relu(x)

def relu2(x):
    return torch.where(x >= 0, x, torch.tensor(0.0))

def max01(x):
    return (x[0] + x[1]) / 2 + relu((x[0] - x[1]) / 2) + relu((x[1] - x[0]) / 2)

def max02(x):
    return (x[0] + x[1]) / 2 + relu2((x[0] - x[1]) / 2) + relu((x[1] - x[0]) / 2)

def max1(x):
    return max01(torch.stack([max01(x[0:2]), max01(x[2:4])]))

def max2(x):
    return max02(torch.stack([max02(x[0:2]), max02(x[2:4])]))

def zero_2(t):
    z = t * x
    return max1(z) - max2(z)
```

Figure 9: Implementation of $\max_1$, $\max_2$ and $\text{zero}_2$ using Pytorch. Programs $\max_1$ and $\max_2$ are an equivalent implementation of max, but implemented using different ReLU-derived programs.

### A.4  NormPool : a nonsmooth multivariate operation without compensation errors

We conducted an experiment to show that compensation errors are not caused by the multivariate nature of nonsmooth elementary functions when using floating-point arithmetic. In this experiment, we used the NormPool operation, which is similar to the MaxPool operation but replaces the

| | | | $\mathrm{zero}_2'(t)$ | | | | | | |
|---|---|---|---|---|---|---|---|---|---|
| $t$ | | | $-10^{-3}$ | $-10^{-2}$ | $-10^{-1}$ | $0$ | $10^1$ | $10^2$ | $10^3$ |
| $x = \begin{vmatrix} 1.0 & 2.0 & 3.0 & 4.0 \end{vmatrix}$ | | | $0.0$ | $0.0$ | $0.0$ | $1.5$ | $0.0$ | $0.0$ | $0.0$ |
| $x = \begin{vmatrix} 1.4 & 1.4 & 1.4 & 1.4 \end{vmatrix}$ | | | $10^{-7}$ | $10^{-7}$ | $10^{-7}$ | $10^{-7}$ | $10^{-7}$ | $10^{-7}$ | $10^{-7}$ |

Table 3: Summary of various types of AD errors with $\mathrm{zero}_2$ program using PyTorch for different combinations of $t$ and $x$.

maximum with the Euclidian norm. Two programs, $P$ and $Q$, were used to implement a LeNet-5 network on the MNIST dataset with two different NormPool-derived programs. We computed the backprop variation (see Definition 7) between $P$ and $Q$, while controlling all sources of divergence in our implementation using deterministic computation. The results are presented in Figure 10. The experiment was conducted on a CPU with 16-bit floating-point precision.

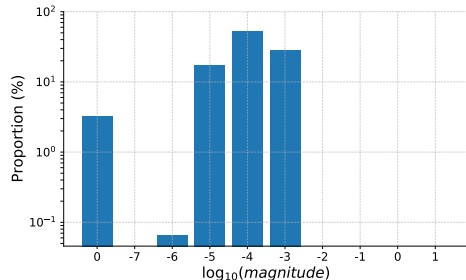

Figure 10: Histogram of backprop variation between $P$ and $Q$ for a LeNet-5 network on MNIST (128 mini-batch size) with 16-bit. We run $M = 1000$ experiments.

In contrast to our findings with MaxPool, we obtained similar results to those reported in Bertoin et al. (2021) with ReLU-based programs. Specifically, for NormPool-based programs, we observed either significant divergence of backprop or none.

## A.5 Bifurcation zone: a practical example

This section presents an example that demonstrates cases where AD can be incorrect. Calculating the accurate derivative for all inputs might be impossible, particularly when the function is nondifferentiable. This is because the derivative does not exist for inputs where the function is nondifferentiable.

### A.5.1 Network configuration

Consider an input matrix $X$ of size $4 \times 4$ given by:

$$X = \begin{pmatrix} 1 & 0 & 0 & 1 \\ 0 & 0 & 0 & 0 \\ 0 & 0 & 0 & 0 \\ 0 & 0 & 0 & 0 \end{pmatrix} \qquad \text{(Input)}$$

Let $k$ be a positive number and $W$ be a convolution kernel of size $3 \times 3$ given by:

$$W = k \cdot \begin{pmatrix} 1 & 1 & 1 \\ 1 & 1 & 1 \\ 1 & 1 & 1 \end{pmatrix} \qquad \text{(Convolution kernel)}$$

Let's consider a composition function $l$ such that:

$$l(W) = \text{MaxPool} \circ (X * W) = k \tag{17}$$

where the convolution operation $X * W$ produces an output matrix $Z$ of size $2 \times 2$, followed by the application of a MaxPool with a pooling window of size $2 \times 2$.

### A.5.2 Backprop computation: native vs minimal

Let $P$ (resp. $Q$) be a program implementing the composition function $l$ in Equation equation 17 using the native (resp. minimal) MaxPool-derived program. Then, we have:

$$\text{backprop}[P](W) = \begin{pmatrix} 1 & 0 & 0 \\ 0 & 0 & 0 \\ 0 & 0 & 0 \end{pmatrix}, \quad \text{backprop}[Q](W) = \begin{pmatrix} 0.5 & 0 & 0.5 \\ 0 & 0 & 0 \\ 0 & 0 & 0 \end{pmatrix}$$

The convolutional kernel $W$ falls within the bifurcation zone defined in Definition 3.

## A.6 Comments on Section 3

### A.6.1 Non-determinism in GPU computation

Graphics Processing Units (GPUs) are designed for parallel processing, which can result in unpredictable behaviors.

- **Floating-point operations:** The non-associative nature of floating-point arithmetic can lead to discrepancies. These differences might become significant as they accumulate across operations.

- **Reduction operations:** Functions like sum or maximum, especially in GPUs, can exhibit variability between runs. This variability can result in divergent accumulated rounding errors.

### A.6.2 Threshold values for various networks in Section 3.1

Table 4 presents threshold values for various neural networks on different datasets, computed under different floating-point precisions (16-bit, 32-bit, and 64-bit). For simplicity, thresholds are approximated as powers of 10.

### A.6.3 Details on Monte Carlo sampling in Section 3.2

Recall that, for a neural network $f$ and a floating-point precision $\omega$, we want to estimate the volume of the set

$$S(\tau_{f,\omega}) = \left\{ \theta \in \mathbb{R}^P : \exists i \in \{1, \dots, N\}, \|\text{backprop}[P_i](\theta) - \text{backprop}[Q_i](\theta)\|_1 > \tau_{f,\omega} \right\} \subset \Theta_B$$

| Network $f$ | Dataset | $\tau^1_{f,16}$ | $\tau^2_{f,16}$ | $\tau^1_{f,32}$ | $\tau^2_{f,32}$ | $\tau^1_{f,64}$ | $\tau^2_{f,64}$ |
|---|---|---|---|---|---|---|---|
| LeNet-5 | MNIST | $0$ | $10^{-5}$ | $10^{-6}$ | $10^{-5}$ | $10^{-14}$ | $0$ |
| VGG-11 | CIFAR-10 | $0$ | $10^{-1}$ | $10^{-8}$ | $10^{-7}$ | $10^{-14}$ | $0$ |
| VGG-11 | SVHN | $0$ | $10^{-1}$ | $10^{-8}$ | $10^{-7}$ | $10^{-15}$ | $0$ |
| VGG-13 | CIFAR-10 | $0$ | $10^{-1}$ | $10^{-9}$ | $10^{-9}$ | $10^{-14}$ | $0$ |
| VGG-16 | CIFAR-10 | $0$ | $10^{-2}$ | $10^{-10}$ | $10^{-9}$ | $10^{-15}$ | $0$ |
| VGG-19 | CIFAR-10 | $0$ | $10^{-3}$ | $10^{-11}$ | $10^{-10}$ | $10^{-15}$ | $0$ |
| ResNet-18 | CIFAR-10 | $10^{-2}$ | $1$ | $10^{-3}$ | $10^{-4}$ | $10^{-13}$ | $0$ |
| DenseNet-121 | CIFAR-100 | $0$ | $10^{-2}$ | $10^{-6}$ | $10^{-1}$ | $10^{-14}$ | $0$ |

Table 4: Threshold values of various neural networks $f$ across different datasets.

Our experiments divide a dataset into $R$ mini-batches. Each $r$-th mini-batch is represented by the index set $B_r \subset \{1, \ldots, N\}$. The programs $P_r$ and $Q_r$ are associated with the neural network $f$ and implement a composition function $l_r$ for each $r$. Specifically, $P_r$ uses the native MaxPool-derived program, whereas $Q_r$ uses the minimal one. For every precision level $\omega \in \{16, 32, 64\}$, we establish a threshold $\tau_{f,\omega}$ as in Section 3. Using the Kaiming-Uniform (He et al., 2015) initialization in PyTorch, we randomly generate a parameter set $\{\theta_j\}_{j=1}^M$, with $M = 1000$. The first line of Table 2 is given by the formula

$$\frac{1}{M} \sum_{j=1}^K \mathbb{1}\left(\exists r \in \{1, \ldots, R\}, \left\|\text{backprop}\left[\sum_{j \in B_r} P_j(\theta)\right] - \text{backprop}\left[\sum_{j \in B_r} Q_j(\theta)\right]\right\|_1 > \tau_{f,\omega}\right), \quad (18)$$

where $\mathbb{1}$ represents the indicator function, returning either 1 or 0 depending on the truth value of its argument's condition. Similarly, the second line of Table 2 is given by the formula

$$\frac{1}{MR} \sum_{j=1}^M \sum_{r=1}^R \mathbb{1}\left(\left\|\text{backprop}\left[\sum_{j \in B_r} P_j(\theta)\right] - \text{backprop}\left[\sum_{j \in B_r} Q_j(\theta)\right]\right\|_1 > \tau_{f,\omega}\right), \quad (19)$$

Using the formula

$$\sqrt{\frac{\ln\left(\frac{2}{\alpha}\right)}{2n}},$$

and setting $\alpha = 0.05$, we compute the error margin of the Hoeffding confidence interval as $n = M$ for Table 2's first line and $n = MR$ for its second. The first line adheres to a 95% confidence interval under the *iid* assumption due to Hoeffding's inequality.

Using McDiarmid's inequality at risk level $\alpha = 0.05$, we compute the error margin of the second line in Table 2 by the formula

$$\sqrt{\frac{1}{2}\left(\frac{1}{M} + \frac{1}{R}\right) \ln\left(\frac{2}{\alpha}\right)}.$$

## B  Proof related to Section 2.3

**Proof 1 (of Proposition 1)**

1. The three subsets have unique definitions, indicating that they are separate. For instance, a parameter cannot belong to the regular and bifurcation zones since the regular zone is defined as the area where each program $g_{i,j}$ is assessed at differentiable points. On the other hand, the bifurcation zone is defined as the region where the set of all possible backprop outputs is not a singleton, indicating non-differentiability at some points. Additionally, the union of these zones covers the entire parameter space $\Theta$ as every parameter must be assigned to one of the three subsets: resulting in differentiable points when evaluated, resulting in nondifferentiable points but having a singleton backprop set, or resulting in nondifferentiable points with a non-singleton backprop set. Therefore, $\Theta_R \cup \Theta_B \cup \Theta_C = \Theta$.

2. As we consider locally Lipchitz semialgebraic (or definable) functions, see [Theorem 1, Bolte & Pauwels (2020a)] for the proof arguments.

## C    Complements on experiments

### C.1    Benchmark datasets and architectures

**Datasets:**    In this work, we utilized various well-known image classification benchmarks. Below are the datasets, including their characteristics and original references.

| Dataset | Dimensionality | Training set | Test set |
|---------|----------------|--------------|----------|
| MNIST | $28 \times 28$ (grayscale) | 60K | 10K |
| CIFAR10 | $32 \times 32$ (RGB) | 60K | 10K |
| SVHN | $32 \times 32$ (RGB) | 600K | 26K |
| ImageNet | $224 \times 224$ (RGB) | 1.3M | 50K |

The corresponding references for these datasets are LeCun et al. (1998); Krizhevsky & Hinton (2010); Netzer et al. (2011).

**Neural network architectures:**    We evaluated various CNN neural network architectures, with details as follows:

| Name | Layers | Loss function |
|------|--------|---------------|
| LeNet-5 | 5 | Cross-entropy |
| VGG11 | 11 | Cross-entropy |
| VGG13 | 13 | Cross-entropy |
| VGG16 | 16 | Cross-entropy |
| VGG19 | 19 | Cross-entropy |
| ResNet18 | 18 | Cross-entropy |
| ResNet50 | 50 | Cross-entropy |
| DenseNet121 | 125 | Cross-entropy |

The corresponding references for these architectures are Simonyan & Zisserman (2014); He et al. (2016); Huang et al. (2017); LeCun et al. (1998).

**LeNet-5:**    The implementation for LeNet-5 was sourced from the following GitHub repository: `https://github.com/ChawDoe/LeNet5-MNIST-PyTorch/blob/master/model.py`.

**VGG:** We used the PyTorch repository's implementation for the VGG models. It can be accessed at the following link: `https://github.com/PyTorch/vision/blob/main/torchvision/models/vgg.py`.

**ResNet:** For ResNet models, we utilized the PyTorch repository's implementation available at: `https://github.com/PyTorch/vision/blob/main/torchvision/models/resnet.py`. We made minor adjustments to the output layer's size (changing from 1000 to 10 classes) and the kernel size in the primary convolutional, varying from 7 to 3). When batch normalization was not used, we replaced the batch normalization layers with identity mappings.

**DenseNet:** The implementation for DenseNet was taken from the PyTorch repository, available at: `https://github.com/PyTorch/vision/blob/main/torchvision/models/densenet.py`.

## C.2 Mitigating factor: Adam optimizer

After training a VGG11 network on CIFAR-10 using the Adam optimizer, we obtained results shown in Figure 11. Our findings are consistent with those presented in Section 3, but the network exhibits reduced sensitivity to $\beta$, resulting in improved stability of both test errors and training loss.

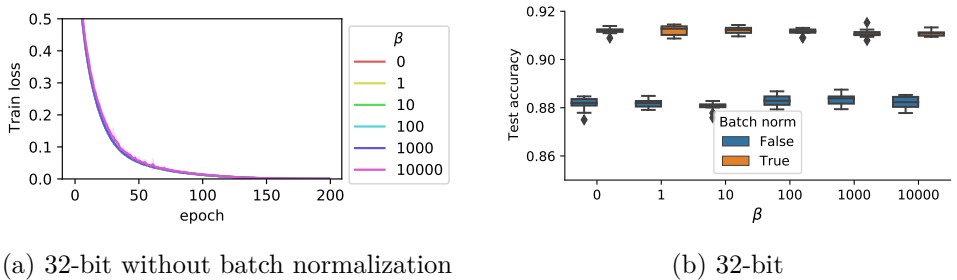

(a) 32-bit without batch normalization          (b) 32-bit

Figure 11: Training losses on CIFAR10 (left) and test accuracy (right) on VGG network trained with Adam optimizer and without batch normalization.

## C.3 Additional experiments with MNIST and LeNet-5 networks

We repeated the experiments in Section 4.2 using a LeNet-5 network on the MNIST dataset. The results are depicted in Figure 12. We found that for 16 bits, the test accuracies were similar when training was possible, but $\beta = \{10^3, 10^4\}$ caused chaotic training behavior. For 32 bits, the test accuracies were mostly similar, except for $\beta = 10^4$. We noticed that the chaotic oscillations had completely disappeared.

## C.4 Additional experiments with ResNet18

We performed the same experiments described in Section 4.2 using ResNet18 architecture trained on CIFAR 10. Figure 13 represents the test errors with or without batch normalization. For 16 bits, test accuracies are similar, but $\beta = 10^4$ induces chaotic training behavior. For 32 bits, test accuracies are identical, and the chaotic oscillations phenomena have entirely disappeared.

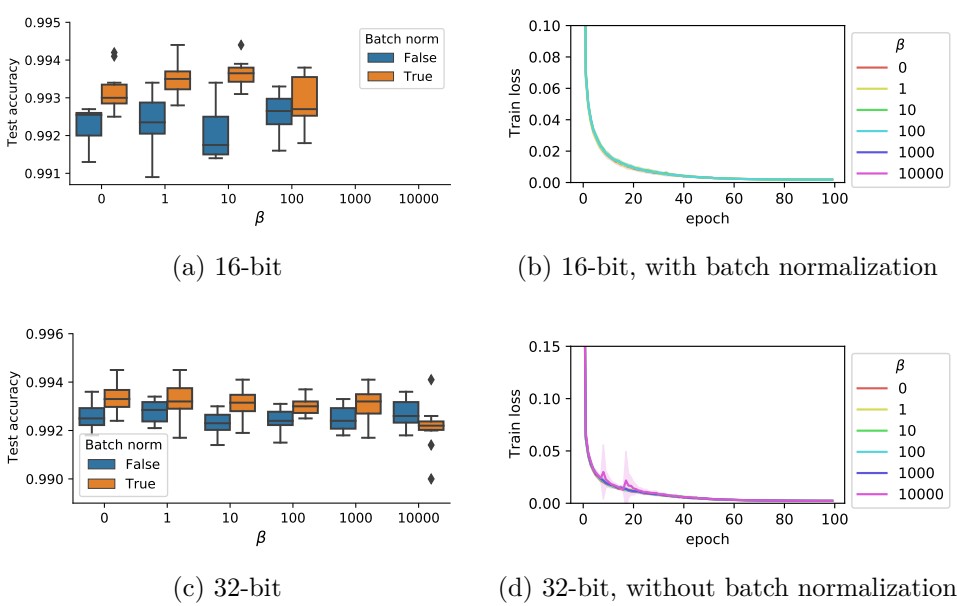

(a) 16-bit

(b) 16-bit, with batch normalization

(c) 32-bit

(d) 32-bit, without batch normalization

Figure 12: Training a LeNet-5 network on MNIST with SGD. We performed ten random initializations for each experiment, depicted by the boxplots and the filled contours (standard deviation).

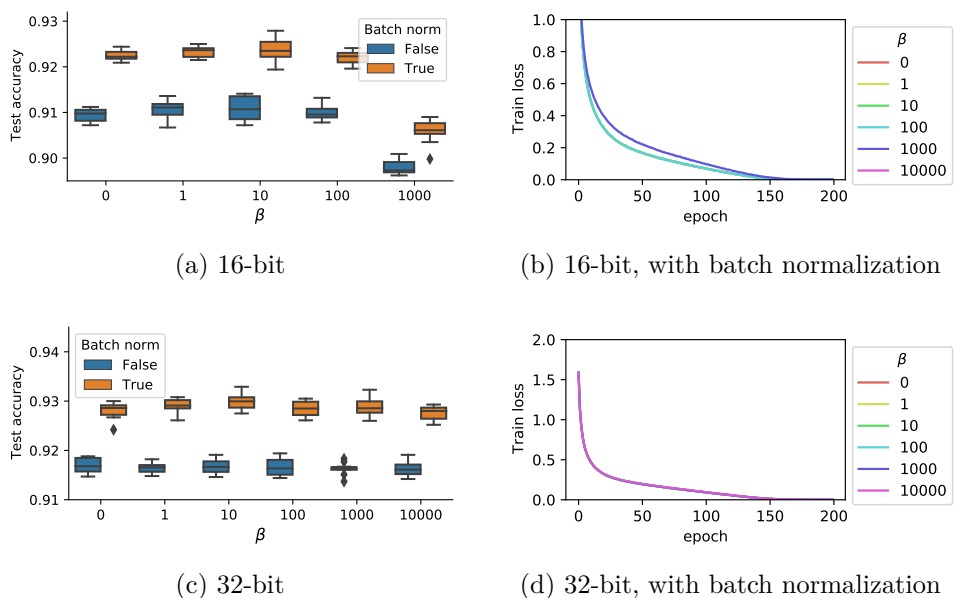

(a) 16-bit

(b) 16-bit, with batch normalization

(c) 32-bit

(d) 32-bit, with batch normalization

Figure 13: Training a ResNet18 network on CIFAR10 with SGD. We performed ten random initializations for each experiment, depicted by the boxplots and the filled contours (standard deviation).

## C.5 Additional experiments with ResNet50 on ImageNet

We performed the same experiments described in Section 4.2 using a ResNet50 architecture trained on ImageNet. The test accuracy is represented in Figure 14. We employ mixed precision (Micikevicius et al., 2017; Jia et al., 2018), utilizing 16 and 32 bits precision to balance computational speed and

information retention. Test accuracies are similar when training is possible, but $\beta = 10^3$ induces chaotic training behavior.

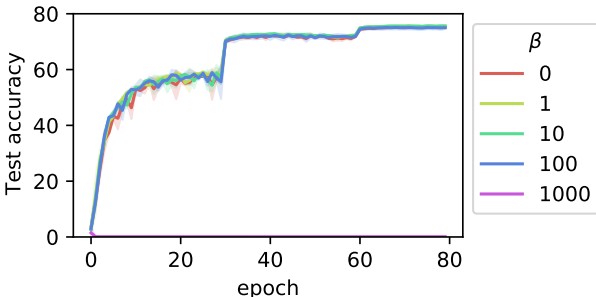

Figure 14: Test accuracy during training a Resnet50 on ImageNet with SGD using mixed precision. The shaded area represents three runs. We have a chaotic test accuracy behavior for $\beta = 10^3$.

## D Complementary information

**Computational Resources:** All the experiments were conducted on four Nvidia V100 GPUs. This ensured consistent and reliable computation times across different experimental runs.

**Code and Results Availability:** The code corresponding to the experiments, as well as the results of these experiments, are publicly available. The repository can be accessed at the following URL: `https://github.com/ryanboustany/MaxPool-numerical`.

**Licenses:** The datasets used in our experiments are released under various licenses. CIFAR10 is under the MIT license, MNIST and SVHN are under the GNU General Public License, and ImageNet is under the BSD license. The libraries we used, Numpy and PyTorch, are released under the BSD license, while Python is released under the Python Software Foundation License.

| Dataset | Network | Optimizer | Batch Size | Epochs | Time Per Epoch | Repetitions |
|---------|---------|-----------|------------|--------|----------------|-------------|
| MNIST | LeNet-5 | SGD | 128 | 100 | 2 seconds | 10 |
| CIFAR10 | VGG11 | SGD | 128 | 200 | 9 seconds | 10 |
| CIFAR10 | ResNet18 | SGD | 128 | 200 | 13 seconds | 10 |
| SVHN | VGG11 | SGD | 128 | 100 | 70 seconds | 10 |
| ImageNet | Resnet50 | SGD | 512 | 90 | 15 minutes | 3 |

Table 5: Detailed experimental setup, including the dataset, neural network architecture, optimizer used, batch size, number of epochs, average computation time per epoch, and repetitions for each experiment.

