# OpenReview forum: "On the numerical reliability of nonsmooth autodiff: a MaxPool case study"
_TMLR — Accepted by TMLR_

### Review · Reviewer_112e · 2024-03-06

**Summary Of Contributions:**

This paper studies inaccuracies in the automatic differentiation of the MaxPool operations. The paper identifies two types of errors: bifurcation errors, occurring in non-differentiable regions of the function, and compensation zones, which are errors that occur purely due to machine imprecision. The authors demonstrate theoretically that the bifurcation zones occur in a set of measure 0. Next, the authors conduct experiments demonstrating when these errors occur and find, among other observations, that higher numerical precision can mitigate these errors somewhat.

**Audience:**

Yes

**Broader Impact Concerns:**

No broader impact concerns.

**Claims And Evidence:**

Yes

**Requested Changes:**

**Would strengthen**

Explaining the implementations of max_1 and max_2 on page 2 instead of in the Appendix would be helpful since this is a key introductory piece of the paper

Increasing the size of some of the figures (especially 5 and 6) for readability

Improving the table format by removing the vertical column dividers

Explaining more clearly the practical significance of the paper's results

**Strengths And Weaknesses:**

**Strengths**

Overall, the paper presents a comprehensive analysis of the differentiability of the MaxPool operation, both from a theoretical and empirical standpoint. Experiments are well-conducted and the presentation of concepts is well-structured. Theoretical results appear correct.

**Weaknesses**

While generally well-written, a few changes could improve the clarity of the manuscript. The proposed changes are outlined below in the requested changes section.

Another weakness is that the practical value and significance of the paper's observations could be made more apparent. The paper analyzes test accuracy for networks with different implementations of MaxPool, but it would be helpful for the paper to make a set of concrete recommendations for practitioners.

---

> ### Author Response · Authors · 2024-05-06
> **Response to Reviewer 112e**
>
> We thank the reviewer for his positive evaluation, comments, and constructive feedback; we provide a detailed answer to his remarks below. Please let us know if you have any follow-up questions.
>
> ### W.1 While generally well-written, a few changes could improve the clarity of the manuscript. The proposed changes are outlined below in the requested changes section.
>
> A.1 Thanks for pointing these out; we will correct all these suggestions.
>
> ### W.2 Another weakness is that the practical value and significance of the paper's observations could be made more apparent. The paper analyzes test accuracy for networks with different implementations of MaxPool, but it would be helpful for the paper to make a set of concrete recommendations for practitioners.
> A.2 It is true; we will discuss concrete recommendations for practitioners.
>
> ### W.3 Explaining the implementations of max_1 and max_2 on page 2 instead of in the Appendix would be helpful since this is a key introductory piece of the paper
> A.3 Thank you for the suggestion. This section will be revised.

---

### Review · Reviewer_rh1s · 2024-04-10

**Summary Of Contributions:**

This paper proposes a study of the influence of the practical implementation of MaxPool on the computed gradients when using AutoGrad, in the context of Deep Learning. The authors propose to distinguish two kinds of error: 1) the floating-point errors are caused by the limited precision when representing reals on computers; 2) the errors related to the choice of implementation of non-smooth functions.

The authors provide an empirical study involving MaxPool and ReLU, with random weights and during training.

**Audience:**

Yes

**Broader Impact Concerns:**

None.

**Claims And Evidence:**

Yes

**Requested Changes:**

Overall, the scope and the clarity of the paper need to be improved.

# Scope

The scope of the paper should be improved: is it about MaxPool? compensation zones? the specific influence of non-deterministic computation? I recommend to deal with MaxPool and ReLU equally, to insist more on the importance of the compensation zones (which seem to be the core of the paper), and thus emphasize the influence of non-deterministic computation.

N.B.: if it is technically irrelevant to study the compensation zones with ReLU, it should be discussed somewhere.

# Clarity

Many definitions lack clarity: $\tau^1$ and $\tau^2$ (which may be both defined with MaxPool or ReLU); use of letters $M$ and $K$; discussion on $\beta > 1$.

One should discuss the relevance of training experiments (which rely on $\beta > 1$). If there is no significant difference between experiments made with any $\beta \in [0, 1]$, it is interesting for the community and should be reported somewhere. I would prefer a discussion about a negative result in a standard setting than about not-so-clear results in an artificial setting (i.e., $\beta > 1$).

**Strengths And Weaknesses:**

# Strengths

## Importance of the problem

This paper explores the influence of the floating-point precision and the practical implementation of MaxPool, and, to some extent, ReLU, on the results of AutoDiff. This kind of study is necessary when an entire community (Deep Learning) relies heavily on AutoDiff. Therefore, this paper tackles a important question.


## Difference between compensation and bifurcation zones

The authors propose to distinguish two kinds of practical limitations: floating-point precision and practical implementation of non-smooth functions. This distinction is fundamental to tackle the initial problem, and is an important contribution of the paper.


# Weaknesses

## Scope of the paper

Overall, the scope of the paper is unclear: it is a focus on MaxPool, but the analysis seems to be feasible with ReLU (and it is partially done), which is more natural to study given the past works.

### Abstract

In the abstract, the distinction between existing works, known results and contributions of the paper is unclear. For instance: are the "bifurcation zone" and the "compensation zone" invented in this paper?

### Intro: Various types of nonsmooth AD errors

In the Introduction, the authors show interesting preliminary results in Table 1. They are presented as the main motivation of the paper: using different implementations of the same function (here, MaxPool) may result in very different gradients.

However, it is explained that these results, obtained with MaxPool, can be replicated with ReLU (see Appendix A.3). Therefore, why do the authors focus on MaxPool (as the title of the paper suggests), and do not claim to do a study involving both MaxPool and ReLU?

### Intro: Related works and contributions

More clearly, the authors explain that they propose the concept of "compensation zone", while the concept of "bifurcation zone" already exists. However, why do they limit their study to MaxPool, while an extension of the preceding works with ReLU would be more natural?

### Section 3.1: $\tau^1$ and $\tau^2$

As far as I understand, informally:
 * $\tau^1$ is the upper bound on the error we expect to make when running several times the same program;
 * $\tau^2$ is the lower bound on the error we expect to make when running two different implementations of the same function.

It is very strange that the authors computed $\tau^1$ only with MaxPool and $\tau^2$ only with ReLU. One may conversely use ReLU in a non-deterministic setting, and test two different implementations of MaxPool.

It is crucial to compute $\tau^1$ and $\tau^2$ with both ReLU and MaxPool: it would help to evaluate the relevance of $\tau^1$ and $\tau^2$ to distinguish the compensation zone and the bifurcation zone in each case.

Besides, within the scope of the paper, it seems very important to be able to run experiments in a "non-deterministic" setting. The importance of this setting should be mentioned in the introduction (if not in the abstract itself).


## 32 bits MNIST experiment (section 3.1)

The authors claim that the results plotted in Figure 1 are significantly different to the ones plotted in [1] (Figure 1).

First, if I understand correctly, results of this section have been obtained with parameters at initialization (i.e., random parameters). However, the results showed in [1] (Figure 1) have been obtained with trained parameters (training experiments). So, these results are not comparable.

Second, even when comparing these plots to the ones presented in the Erratum of [1] (Figure 1), one should note that the weight difference is between $0$ and $10^{-3}$ in both cases. So, the "contrast" emphasized by the authors with the results of [1] is not very striking.

I recommend to run experiments in the same condition to test the numerical stability of AD with ReLU and MaxPool: either both with random weights, or both throughout training.


## $\mathrm{P}$ versus $\tilde{\mathrm{P}}$ in a deterministic setup

Since $\tilde{\mathrm{P}}$ is a copy of $\mathrm{P}$ and all the computations are deterministic, I do not understand how experiments with $\mathrm{P}$ and with $\tilde{\mathrm{P}}$ could lead to different results. Thus, Figure 1.b is superfluous (or could be, at most, regarded as a sanity check).


## Clarity

### Representation of a neural network: Eqn. (2)

Even though it looks familiar, the representation of a neural network (NNs) proposed in Eqn. (2) is very uncommon. This kind of representation is also used in [1], but it comes with a short appendix explaining how it works. I recommend the authors to add a short description (possibly in the appendix) explaining how to represent usual NNs in this form.

### Notation $M$ used twice

The letter $M$ is used as a number of layers (Section 2) and as a number of experiments (Section 3). For clarity, I recommend to choose a different letter.

Besides, the letter $K$ is also used as a number of experiments (Figure 2).

### $\beta$: what is the meaning of $\beta > 1$?

$\beta$ is used as a weight in the "hybrid" MaxPool (Definition 6). The authors do not explain in-text why we should be insterested in values $\beta > 1$. Since most of experiments are made with $\beta > 1$, the reason of this choice should be explained in this paper.

### Definition of $\tau^1$ and $\tau^2$

As far as I understand, informally:
 * $\tau^1$ is the upper bound on the error we expect to make when running several times the same program;
 * $\tau^2$ is the lower bound on the error we expect to make when running two different implementations of the same function.

They are two distinct ways to delimit the compensation set ("by above") and the bifurcation set ("by below").

But this is an inference on my part (and I am not sure to understand correctly). To improve clarity, I recommend to:
 * provide an intuition of what the authors intend to measure with $\tau^1$ and $\tau^2$;
 * define first $\tau^1$ and $\tau^2$ on *only one* function (ReLU or MaxPool, it does not matter), then extend it to the other; as such, one may think that $\tau^1$ is defined with MaxPool and $\tau^2$ with ReLU.


## Important typos

 * Figure 4, middle, label of x-axis: write "Mini-batch size" instead of "Number of mini-batch size" (which is unclear);
 * Appendix A.6.3: the indicator function is badly typeset; "$\mathbb{1}$" should be used;
 * Section 4.2: "$\beta \in \\{ 1, 10, 100\\}$" instead of "$\beta \in 1, 10, 100$".


## Reference to *Numerical influence of ReLU'(0) on backpropagation*

Apparently, [1] has been accepted as a NeurIPS poster in 2021. Why is this information missing in the References section (with 2023 as year of publication)? If the authors want to refer to the *Erratum* of [1], this should be mentioned in the information about [1] in the References section (by adding "Erratum" in the title of the paper, for instance). Otherwise, this reference may refer to two distinct works.

Besides, even if it is worth mentioning the Erratum of [1], it would be better to refer to [1] directly for results that are not impacted by the error present in [1].

[1] *Numerical influence of $\mathrm{ReLU}'(0)$ on backpropagation*, Bertoin et al., 2021.

---

> ### Author Response · Authors · 2024-05-06
> **Response to Reviewer rh1s (1/2)**
>
> We thank the reviewer for his positive comments and constructive feedback. Comments are answered in detail below. We believe they addressed the reviewer's concerns about the paper’s scope and clarity.  Please let us know if you have any follow-up questions.
>
> ### W.1 Overall, the scope of the paper is unclear: it is a focus on MaxPool, but the analysis seems to be feasible with ReLU (and it is partially done), which is more natural to study given the past works.
> A.1 Yes, we understand, and we will revise our scope to convey more clearly. Bertoin et al. studied the ReLU'(0) effect on AD and training, identifying a bifurcation zone in ReLU networks: the set of the network parameters on which the output of AD using ReLU’(0)=0 is different from that using ReLU’(0)=1. However, they do not observe numerically cases where AD is incorrect over floating-point numbers but corrects over real numbers. To observe this phenomenon, we need to compute MaxPool with different derivative implementations.
> Although it is possible to calculate the max using the ReLU operation, the compensation zone arises from the computation of MaxPool autodiff with varying derivatives. Altering the value of ReLU’(0) while calculating the max can result in a different implementation of max derivatives. Thank you for your feedback, we will revise the text to make it clearer.
>
> ### W.2 In the abstract, the distinction between existing works, known results and contributions of the paper is unclear. For instance: are the "bifurcation zone" and the "compensation zone" invented in this paper?
> A.2 The numerical bifurcation zone proposed by Bertoin et al. does not consider the case where AD is incorrect over floating-point numbers but correct over real numbers. Hence, we propose a new numerical bifurcation zone to identify two types of errors: bifurcation errors, occurring in non-differentiable regions of the function, and compensation zones, which are errors that occur purely due to machine imprecision.
>
> ### W.3 However, it is explained that these results, obtained with MaxPool, can be replicated with ReLU (see Appendix A.3). Therefore, why do the authors focus on MaxPool (as the title of the paper suggests), and do not claim to do a study involving both MaxPool and ReLU?
> A.3 The numerical bifurcation zone identified by Bertoin et al. does not apply to MaxPool-derived program analysis. As indicated in Table 1, our investigation includes AD errors that occur with floating-point but not with real numbers. As the MaxPool function is not implemented in practice with ReLUs, we decided to study autodiff on the native implementation of MaxPool. Thank you for the question; we will add a remark about this.
>
> ### W.4 More clearly, the authors explain that they propose the concept of "compensation zone", while the concept of "bifurcation zone" already exists. However, why do they limit their study to MaxPool, while an extension of the preceding works with ReLU would be more natural?
> A.4 See A.1 and A.3.
>
> ### W.5 It is very strange that the authors computed $\tau^1$ only with MaxPool and $\tau^2$ only with ReLU. One may conversely use ReLU in a non-deterministic setting and test two different implementations of MaxPool.
> A.5 To identify the numerical bifurcation zone (occurring in non-differentiable regions of the function), we have to find criteria to avoid errors that occur purely due to machine imprecision. Hence, we compare observed backprop variations with known sources: GPU non-determinism ($\tau^1$) and the AD variations from ReLU-derived programs ($\tau^2$), which are null or above the machine precision.
>
> $\tau^1$ : We compute $\tau^1$ for a LeNet-5 network, which is composed by MaxPool and ReLU operations. So $\tau^1$ is computed with MaxPool and for ReLU. Thank you, we will clarify this in the paper :  changing « using the same MaxPool derived-program » by « using nonsmooth derived-program (for e.g ReLU or MaxPool) ».
>
> $\tau^2$: As shown in Bertoin et al., we don't always have AD variation. If we do have a variation, it's well above machine precision (refer to Figure 3): this is another criteria for the numerical bifurcation zone.
>
> ### W.6 32 bits MNIST experiment (section 3.1)
> A.6 In Bertoin et al. (Figure 1), the authors do not observe any weight difference until a certain iteration of training. You are right; it is not the same experiment in our paper. We will correct this sentence.
> In our paper, we run experiments in the same condition (random weights) to test the numerical stability of AD with ReLU and MaxPool. For ReLU-derived programs (Figure 3), we observe no weight difference or weight difference above the machine precision.
>
> ### W.7 $P$ versus $\tilde{P}$ in a deterministic setup.
> A.7 The reviewer is perfectly right. Figure 1.b is only a sanity check, and experiments $P$ versus $\tilde{P}$ can not yield different results.

---

> ### Author Response · Authors · 2024-05-06
> **Response to Reviewer rh1s (2/2)**
>
> ### W.8 Representation of a neural network: Eqn. (2)
> A.8 It is true; we will add a short description or refer to the Appendix used in [1].
>
> ### W.9 Notation $M$ used twice
> A.9 Thank you, we will correct this.
>
> ### W.10 $\beta$: what is meaning of $\beta$>1:
> A.10 Thank you again for this interesting and informative question. In Remark 5, we mentioned that $\beta$>1 corresponds to a selection of a conservative Jacobian. The chain rule, essential for AD, often fails with Clarke subgradients, and the Backprop set (Definition 2) is not always a Clarke subdifferential. Thus, we decide also to study conservative Jacobian ($\beta>1$) as they bear a variational sense with nonsmooth autodiff.
>
> ### W.11 Definiton of $\tau_1$ and $\tau_2$
> A.11 Thank you for this comment; we will clarify it in the new version.
>
> ### W.12 Important typos
> A.13  Thanks for pointing these out, we will implement all these suggestions.
>
> ### W.13 Reference to Numerical influence of ReLU'(0) on backpropagation
> A.13 The reviewer is perfectly right. We will correct it.

---

> > ### Comment · Reviewer_rh1s · 2024-05-13
> >
> > Overall, I am satisfied with the answer of the authors. I disagree slightly with Reviewer 7NMi: as such, the paper studies how numerical instabilities can happen. To me, the study is significant enough for TMLR.
> >
> > However, since the revision of the authors is not available, I cannot check if the scope and clarity concerns have been properly addressed.

---

> > > ### Author Response · Authors · 2024-05-15
> > > **Response to Reviewer rh1s (3)**
> > >
> > > Thanks for your timely and instructful feedback! We have uploaded an updated version of the paper that reflects the feedback and changes.

---

### Review · Reviewer_7NMi · 2024-05-05

**Summary Of Contributions:**

The authors present the analysis of the (sub)gradient computation for the MaxPool layer in the classical convolutional neural networks. The main contribution of the presented manuscript is the introduction of two separate groups of parameters corresponding to bifurcation and compensation zones. These zones indicate the large and small (of the machine precision order) deviations of gradient elements from the expected values, respectively. Extensive numerical experiments with CNNs demonstrate the changes in gradient computation if 16-, 32-, and 64-bit floating point numbers are used to store the CNNs' parameters. Also, the authors highlight the importance of the well-known BatchNorm layer in mitigating instabilities obtained from the large-norm Jacobian in the MaxPool layer.

**Audience:**

Yes

**Broader Impact Concerns:**

No concerns on the ethical implications.

**Claims And Evidence:**

No

**Requested Changes:**

1) focus on the suggestions on how to modify the MaxPool backward step to improve stability (improve the test accuracy) or make this step adaptive for different floating point formats.
2) add the same analysis for bfloat16 af tensorfloat32 formats since they are specially developed for training neural networks
3) improve the quality of the plots and make the conclusions from them explicit
4) revise the section about the experiments' results to make it consistent, and introduce modifications to the training setup incrementally to observe the effect of a particular ingredient. The explicit conclusions on the impact of Adam, the BatchNorm layer, and the proper computing of the MaxPool gradient make the results more convincing and easy to follow.

**Strengths And Weaknesses:**

Strengths
1) the manuscript is well-written, and the motivation of the study is clear
2) the suggested concepts look interesting and promising for analysis of the instability in training NN
3) the numerical experiments with different floating-point formats illustrate the introduced theoretical concepts

Weaknesses
1) the main weakness of the presented work is the artificially constructed experimental setup. In particular, the learning rate equals 1, which is quite a large value; the tested values of \beta look unnatural since they are too large to be used in practice. So, the authors in experiments operate with the possible scenarios which, however, do not typically appear in the workflow
2) as far as I know, there are no issues in training the discussed CNNs for standard benchmarks via the most popular frameworks, e.g., PyTorch or Jax. So, the instability phenomenon of the gradient computations for the MaxPool layer is important yet does not significantly affect the final performance in the recommended experimental setup. At least, the authors do not demonstrate the possible improvement in the experiments (see section below)
3) the authors focus only on the VGG-type CNN in the main text, although ResNet and DenseNet also include MaxPool layers. Thus, the universality of the discussed issues and suggested remedies is not convincing.
4) the impact of the manuscript on using the MaxPool layer is limited since the authors do not present any recipes for modifying the standard implementation of the MaxPool layer in frameworks.

Also, there are multiple typesetting issues; see the list below.
1) captions to the tables should be placed above, not below
2) double the "equation" word in the caption to Figure 4 and refer to the equation from the appendix, which seems strange. Why not just move this equation into the main text?
3) in most plots, lines for different betas are indistinguishable
4) probably, \cdot command is more appropriate rather than \times in the Hybrid MaxPool program
5) Figure 4 shows the 64-bit precision in legend, but the corresponding color is absent on the histogram

---

> ### Author Response · Authors · 2024-05-06
> **Response to Reviewer 7NMi**
>
> We thank the reviewer for his comments and constructive feedback. Comments are answered in detail below. Please let us know if you have any follow-up questions. We take into account the requests for changes and will update them in a new version.
>
> ### W.1 The main weakness of the presented work is the artificially constructed experimental setup. In particular, the learning rate equals 1, which is quite a large value; the tested values of \beta look unnatural since they are too large to be used in practice. So, the authors in experiments operate with the possible scenarios which, however, do not typically appear in the workflow.
>
> A.1 We use a learning rate of 0.1 for SGD with momentum on CIFAR10, as reported in the literature. Sorry for the typo. We will correct it in the new version.
>
> The choice of Maxpool Jacobian for training NN is one of our results but does not represent the main part.
> For reasonable choices of $\beta$, we do not observe significant effects regarding training loss or test accuracy. We observe an effect only for very large  $\beta$ values, which do not correspond to practical scenarios but have a variational sense. See also W.10 in Response to Reviewer rh1s. We will add a discussion about this.
>
> ### W.2 As far as I know, there are no issues in training the discussed CNNs for standard benchmarks via the most popular frameworks, e.g., PyTorch or Jax. So, the instability phenomenon of the gradient computations for the MaxPool layer is important yet does not significantly affect the final performance in the recommended experimental setup. At least, the authors do not demonstrate the possible improvement in the experiments (see section below).
> A.2 From our knowledge, it’s true that the native MaxPool implementation ($\beta=0$) used in practice has no issues in training, even using 16 bits (where the bifurcation zone thickens the most). We believe that is an interesting result for the community: even if AD is incorrect with MaxPool, training remains stable.
>
> ### W.3 The authors focus only on the VGG-type CNN in the main text, although ResNet and DenseNet also include MaxPool layers. Thus, the universality of the discussed issues and suggested remedies is not convincing.
>
> A.3 In Appendix C, we show the results for LeNet-5 on MNIST, ResNet-18 on CIFAR10, and ResNet-50 on ImageNet. We obtain qualitatively similar results to those of VGG-type networks. Thank you, we will discuss this more precisely in the main text.
>
> ### W.4 The impact of the manuscript on using the MaxPool layer is limited since the authors do not present any recipes for modifying the standard implementation of the MaxPool layer in frameworks.
>
> A.4 We show that using a MaxPool-derived program with a lower norm enhances training stability and test accuracy, and in particular, we focus on the native ($\beta=0$ used in practice) and the MaxPool-derived program with the lowest norm (our implementation with $\beta=1$).  Although the experiments performed with $\beta \in [0,1]$ did not show significant differences (on test accuracy), we still believe the results are interesting for the community. We do not see this as a simple/minor result.
>
> ###  W.5 Typesetting issues
>
> A.5 Thanks for pointing these out; they will of course be corrected in the revision.

---

> > ### Comment · Reviewer_7NMi · 2024-05-08
> > **Response to the authors**
> >
> > Dear authors,
> > Thanks for the detailed response!
> > Now, I see that your submission primarily describes existing phenomena rather than recommends revision of the current MaxPool backward implementation. The main issue with this approach is that you do not have a specific problem you want to solve since the current implementations work fine. If you focus on the description of the MaxPool backward variabilities, then probably the better research direction is to analyze other non-smooth functions used in deep neural networks from this perspective, too. Otherwise, you consider the single case (MaxPool) and conclude that the standard implementation is ok. So, what can a reader find in your work that he/she can use in his/her own practice?
> > At the same time, a comprehensive survey on whether all popular non-smooth functions are robust to the choice of the subgradient can be interesting. And if some of them are sensitive, can we use this property to improve training quality?
> > Thus, I think a significant paper revision is needed, although the preliminary results look interesting.

---

> > > ### Author Response · Authors · 2024-05-09
> > > **Response to Reviewer 7NMi (2)**
> > >
> > > Dear reviewer,
> > >
> > > Thank you for taking the time to provide us with your comments and constructive feedback. We have carefully considered your feedback and have addressed your concerns in detail below. We are working on a new version of the paper, and we will make improvements to the scope of the paper.
> > >
> > > ### 1. The main issue with this approach is that you do not have a specific problem you want to solve since the current implementations work fine. If you focus on the description of the MaxPool backward variabilities, then probably the better research direction is to analyze other non-smooth functions used in deep neural networks from this perspective, too.
> > >
> > > Firstly, we aim to improve the knowledge about nonsmooth AD correctness [3,5] . The correctness of AD has been extensively studied for decades and for several non-smooth functions used in deep neural networks (ReLU [2], pointwise function [4], ReLU networks with bias [4]). However, it lacks knowledge of where AD is incorrect over floating-point numbers but correct over real.
> > >
> > > Lee et al. [4] studied the correctness of AD over floating-point inputs, proving that it is close to zero for many neural networks. However, their findings may not extend to many practical neural network applications, particularly neural networks employing MaxPool. Furthermore, although Bertoin et al. [2] explored the bifurcation zone (where AD is incorrect over real numbers) for neural networks with ReLU activations, their analysis does not cover networks that use MaxPool operations.
> > >
> > > As a lot of research was done for ReLU networks [2] and pointwise neural networks [4], we focus on the MaxPool operation to extend the previous works of AD correctness:
> > >
> > > 1. We introduced a new subset of weight parameters: the compensation zone (where AD is incorrect over floating-point numbers but correct over reals).
> > >
> > >
> > > 2. We propose a new numerical bifurcation zone that works for all nonsmooth operations in convolutional neural networks (ReLU and MaxPools).
> > >
> > > Overall, we believe this extension is an important addition to a line of recent works (e.g. [2,3,4]) on understanding the effects of selecting proxy gradients that have not yet been considered.
> > >
> > > ### 2. Otherwise, you consider the single case (MaxPool) and conclude that the standard implementation is ok. So, what can a reader find in your work that he/she can use in his/her own practice? At the same time, a comprehensive survey on whether all popular non-smooth functions are robust to the choice of the subgradient can be interesting. And if some of them are sensitive, can we use this property to improve training quality?
> > >
> > > Bertoin et al. (2021) have shown that ReLU’(0) can be adjusted to improve training quality and test accuracies. They discovered that ReLU’(0)=0, which has the minimal norm, provides the best performance. As a result, our aim was to enhance the numerical correctness of AD with MaxPool and introduce a new MaxPool Jacobian implementation using the minimal norm ($\beta=1$). In our paper, we discovered that selecting reasonable values for the parameter $\beta$ did not significantly impact the training loss or test accuracy. Interestingly, Bertoin et al. recently published an erratum to their paper, which supports our findings. They also observed that selecting reasonable values for the parameter ReLU’(0) did not impact the training loss or test accuracy.
> > >
> > > Our paper and the new erratum of Bertoin et al. provide valuable insights for practitioners who are interested in the practical consequences, training, learning, and impact of nonsmooth derivatives. We have found that non-smooth functions for CNN are highly robust to a reasonable choice of subgradient. Through numerical verification, we also have discovered that subgradients with minimal norms are the most stable for training convolutional neural networks.
> > >
> > > ### References
> > >
> > > [1] Sham M. Kakade and Jason D. Lee. Provably correct automatic sub-differentiation for qualified programs. In Annual Conference on Neural Information Processing Systems (NeurIPS), pages 7125–7135, 2018.
> > >
> > > [2] David Bertoin, Jerome Bolte, Sébastien Gerchinovitz, and Edouard Pauwels. Numerical influence of ReLU’(0) on backpropagation. In Annual Conference on Neural Information Processing Systems (NeurIPS), pages 468–479, 2021.
> > >
> > > [3] Wonyeol Lee, Hangyeol Yu, Xavier Rival, and Hongseok Yang. On correctness of automatic differentiation for non-differentiable functions. In Annual Conference on Neural Information Processing Systems (NeurIPS), pages 6719–6730, 2020.
> > >
> > > [4] Wonyeol Lee, Sejun Park, and Alex Aiken. On the correctness of automatic differentiation for neural networks with machine-representable parameters. In International Conference on Machine Learning (ICML), pages 19094–19140, 2023.
> > >
> > > [5] Jérôme Bolte and Edouard Pauwels.Conservative set valued fields, automatic differentiation, stochastic gradient methods and deep learning.Mathematical Programming, pages 1–33, 2020.

---

### Author Response · Authors · 2024-05-15
**Common response to all reviewers**

We thank all reviewers for their comments. We have posted a revised version of the document which we hope represents a significant improvement and addresses the concerns raised, in conjunction with our responses here. We have indicated in the specific responses from reviewers where the relevant revisions have been made. All changes to the document are highlighted in blue. Please let us know if you have any follow-up questions.

---

### Decision · Action_Editor_XHJu · 2024-06-12

**Recommendation:** Accept as is

**Comment:**

This paper studies two types of numerical errors that can occur when training a neural network involving ReLU and MaxPool operators, i.e., floating-point errors by the limited precision on computers and the errors on the implementation of non-smooth functions. The paper introduces theoretical concepts on the differentiability of the MaxPool operation, which are also verified by extensive experiments. The paper is clearly written. The authors have made the changes suggested by the reviewers. All the reviewers are satisfied with the revision.

**Audience:**

This paper explores the influence of floating-point precision and the practical implementation of MaxPool in the results of AutoDiff. This study would be beneficial to the deep learning community since AutoDiff is widely used in the community.

**Claims And Evidence:**

The paper studies an important problem about the implementation of MaxPool in AutoDiff. The paper presents well-conducted experiments, which illustrate the theoretical concepts introduced in the paper. All the reviewers appreciate the extensive empirical verification.